complexity/computer modelling and simulation/mathematical modelling

crypto assets, adoption, cryptocurrencies

**Author for correspondence:**
Silvia Bartolucci
e-mail: s.bartolucci@imperial.ac.uk

# A model of the optimal selection of crypto assets

Silvia Bartolucci[1] and Andrei Kirilenko[2,3]

[1]Department of Finance, Imperial College Business School, London SW7 2AZ, UK
[2]Department of Finance, Cambridge Judge Business School, Cambridge CB2 1AG, UK
[3]Centre for Economic Policy Research, London EC1V 0DX, UK

SB, 0000-0003-1127-5600

We propose a modelling framework for the optimal selection of crypto assets. We assume that crypto assets can be described according to two features: *security* (technological) and *stability* (governance). We simulate optimal selection decisions of investors, being driven by (i) their attitudes towards assets' features, (ii) information about the adoption trends, and (iii) expected future economic benefits of adoption. Under a variety of modelling scenarios—e.g. in terms of composition of the crypto assets landscape and investors' preferences—we are able to predict the features of the assets that will be most likely adopted, which can be mapped to macro-classes of existing crypto assets (stablecoins, crypto tokens, central bank digital currencies and cryptocurrencies).

## 1. Introduction

A crypto asset is an intangible digital asset whose issuance, sale or transfer are secured by cryptographic technology and shared electronically via a distributed ledger. A distributed ledger, in turn, is a database of issuance and transaction records (ledger), copies of which are stored on multiple computing devices (nodes) that form a distributed computer network. A well-known instance of a distributed ledger is the *blockchain*, where information about transactions is stored in a characteristic format consisting of blocks of data chained together via cryptographic primitives. In the following, we will use the two terms 'blockchain' and 'distributed ledger' interchangeably for simplicity.

Each crypto asset has its own blockchain supported by its own network of nodes that provide processing power and memory capacity. All nodes of a given blockchain store copies of the entire history of the blockchain. Some nodes also support a blockchain by selecting, validating and adding/chaining new blocks of records to the ledger in accordance with a pre-specified (consensus) algorithm [1–3].

Starting with the issuance of Bitcoin on the Bitcoin blockchain in 2008 [4], thousands of crypto assets have been issued over the last decade.

The crypto asset ecosystem currently encompasses assets with very diverse underlying technological features (specific cryptographic technologies and electronic sharing protocols), as well as varying governance solutions (private versus open access to the ledger and a variety of consensus algorithms). Vitalik Buterin, the creator of Ethereum, describes a blockchain as 'a magic computer that anyone can upload programs to and leave the programs to self-execute, where the current and all previous states of every program are always publicly visible, and which carries a very strong cryptoeconomically secured guarantee that programs running on the chain will continue to execute in exactly the way that the blockchain protocol specifies' [5].

Reasons for investing in a crypto asset as a *cryptoeconomically secured guarantee of stable execution of a computer at a future date* may substantially differ from reasons for investing in existing assets like stocks, bonds or commodities. Moreover, under the current standards, a crypto asset does not meet the definition of either cash or financial instrument because it does not represent a claim or contractual relationship that results in a monetary or financial liability on any identifiable entity [6]. A crypto asset, however, is an intangible digital asset, as it is without physical substance (intangible), but is digitally identifiable (digital) and held in expectation of future economic benefits (asset). Finally, similarly to other assets, returns on individual crypto assets have been observed to be related to the aggregate return on the 'market' for the asset class as a whole [7].

As such, crypto assets have been sold by issuers ranging from genuine to outright fraudulent, bought by scores of investors with different degrees of sophistication and, consequently, attracted the scrutiny of regulators worldwide. While the majority of crypto assets will eventually become worthless, some could end up being adopted widely enough to ensure their survival. Furthermore, a very small number of crypto assets could—despite likely booms and busts in their prices—become preferred assets of the future and used by large and small investors alike to store and transfer wealth in a cryptographically protected, intangible, digitally native form.

How are investors going to be making optimal selection decisions over many available crypto assets? Which features of these intangible digital assets would drive investment choices? Which types of assets (that do not represent on-blockchain liabilities) will survive and which will go extinct?

In this paper, we propose a modelling framework for the selection among existing and future crypto assets. In our framework, crypto assets can be categorized according to two essential features: *security* and *stability*. Security of a crypto asset represents the technological sophistication of the cryptographic and electronic communication technologies used to withstand cyber fraud, manipulation, abuse and attack. Use of a more advanced encryption technology would render a crypto asset more secure relative to other crypto assets. Stability of a crypto asset reflects its vulnerability to internal attacks that take advantage of the *structure, governance and network architecture* of its blockchain. Use of credible legal, regulatory and self-regulatory (e.g. consensus mechanism) attributes makes a crypto asset more stable.

We assume that demand for crypto assets is driven by the probabilities of their adoption, i.e. investments in a given asset. We can describe the process of crypto assets selection by investors via the following analogy. We posit that investors communicate their preferences for investing in specific crypto assets to a digital platform, defined in the following as 'crypto app'. The app collects data about suggested preferences from all investors, calculates the overall state of the 'market' to date, and then provides investor-specific recommendations.

Nowadays, there are numerous 'digital marketing platforms' used to promote products and services [8]. Familiar examples in the context of 'digitally native (non-financial) assets' include e-books, movies and music records recommender apps. A recommender app or platform can be defined as a type of filtering system that aggregates information (similarly to Internet search engines) and provides user-specific recommendations based on the revealed preferences of all of its users to date. In the context of crypto-investments and wealth management, for instance, interesting examples are provided by the following platforms. On *Moneyfarm* (https://www.moneyfarm.com/uk/investment-advice/), users answer questionnaires to identify their attitude to risk, investment requirements and financial history, and the 'robo-advisor' service provides advice on digital investments. On *Etoro* (https://www.etoro.com), users can trade currency pairs, indices and commodities via the platform: on this app, individuals can also peek into other users' profiles and literally 'copy' their actions, portfolio and preferences and embed them into their own investment strategy.

In our model, the app presents each investor with a pair of crypto assets with certain security–stability characteristics to be compared; each investor submits their preference for adopting one of the two assets to the app; the app, in turn, provides a recommendation on whether the proposed adoption is sensible given the assets' essential features, information about the adoption choices of all other investors, and expected future economic benefits of adoption. Investors continue making their adoption choices over all pairs of crypto assets until their expected future economic benefits can no longer be improved upon, which

**3**

embodies the paradigm of 'optimal selection' decisions. We simulate investors' optimal selection decisions on a landscape composed of crypto assets with different security and stability features.

## 1.1. Related works

Our paper contributes to the emerging literature on crypto assets and blockchains. One strand of the literature studies the features of different crypto assets classes with the aim to categorize them and predict their future behaviour. For example, in [9], the authors present a universal taxonomy to navigate and distinguish different blockchains and distributed ledgers according to the building blocks and subcomponent of the platform (e.g. cryptography, consensus mechanism, etc.). In [10], instead, crypto tokens and cryptocurrencies are classified according to their market capitalization: the power law distribution of the crypto tokens market capitalization presents a larger exponent compared to cryptocurrencies. The growth of the two classes of currencies and their market capitalization growth and distribution were described via an analogue of a population dynamics model.

Another strand of the literature studies economic incentives embedded in blockchain platforms, with particular reference to Bitcoin. In [11–13], the incentive mechanism of mining fees and the costs associated with mining activities are investigated. In [14–16], modelling and analysis of cryptocurrencies incentives is performed and optimal cryptocurrencies designs are proposed. In our model, we do not specifically discuss and include low-level technological features of the platform (e.g. consensus or encryption protocols) but we provide a macro-classification of the assets in terms of two main attributes, security and stability.

Among the theoretical models investigating the mechanisms of acceptance of new digital currencies versus standard (government-issued) currencies, it is worth mentioning the work in [17]. The authors show—by using simple models of standard currency adoption, in particular [18]—how network effects and switching costs may hinder the adoption of a new currency even if all agents agree on its superiority compared to existing ones. In our paper, we consider adoption and switches between digital-only currencies (i.e. crypto assets), for different types of investors, modelling also the impact on returns.

Studies using empirical data focus on understanding the dynamics of the cryptocurrency market using machine learning techniques [19,20], exploring the properties and inefficiencies of the Bitcoin peer-to-peer network [21] and the evolution and categorization of the network of payments on the Bitcoin blockchain [22]. Empirical researches also include cryptocurrencies price forecast and valuation [23] and investigations of the structure, efficiency and maturity of the cryptocurrency markets [24–27]. Agent-based models of cryptocurrency markets have also been simulated to understand the functioning and peculiarities of the real markets. For example, Cocco *et al.* [28] present a Bitcoin market with different types of traders (chartist and/or random), reproducing some key stylized facts of the market, such as Bitcoin price typical time series and volatility clustering. Several studies are also investigating the drivers of the volatility of major cryptocurrencies, showing a strong link between the volatility and the global economic activity [29,30] and a weak dependence on country-specific factors [30].

A crucial role in understanding the cryptocurrency price fluctuations is played by sentiment analysis [31–34], analysis of Internet trends (e.g. via Google trends) and searches to understand the interplay between sentiment extracted from news (e.g. tweets), information available on the Internet and prices [35,36]. Several socio-economic signals, such as volume of word-of-mouth communications and number of new adopters, appears to be intertwined with price dynamics and especially with price movements [37,38]. In our analysis as well, we will focus on and model the interplay between network effects (i.e. an increase in number of investors of an asset may increase the value of the same asset as perceived by other users), adoption and returns. The above-mentioned research strand provides an overview on the type of information that can be easily gathered regarding crypto assets, and how it can be processed and analysed to empirically quantify the effects on adoption and returns.

The paper is organized as follows. In §2, we introduce the crypto assets classification framework, describing the decision-making process of investors for crypto assets adoption. In §3, we show the simulations results for the case of homogeneous investors and heterogeneous investors. Finally, in §4, we discuss possible extensions and future outlooks.

## 2. Methods

In this section, we introduce the details of the model underlying the simulations of the dynamics of crypto assets' returns and adoption. The modelling is framed, for descriptive purposes, within the

context of a digital platform, whereby the demand and supply of crypto assets available to investors is optimized via a 'crypto app'.

## 2.1. Supply of crypto assets

On the supply side, there are $N \gg 1$ crypto assets made available to investors. We assume that crypto assets differ from each other by two essential features: *security* and *stability*. These features reflect two principal directions from which a crypto asset can be attacked and compromised—namely external and internal (to the blockchain).

Security of a crypto asset represents its *technological* vulnerability to an attack that exploits cryptographic primitives employed by the asset holders to prove their credentials and sign their transactions. There is no third-party trusted authority to verify identities and authorize transactions on a blockchain. Holders of a crypto asset must use a public-private key encryption technology to do that. The encryption technology that generates private keys can be vulnerable to an attack if, for example, there is insufficient randomness embedded in the signature process [39]. If a private key can be uncovered and then used to create a digital signature to transfer assets, there is no way for the previous owner to recover the assets. Security features of an asset make their owner more or less vulnerable to a risk that their wallet, including the one held at a service provider like a crypto exchange, is digitally pick-pocketed. Use of a more advanced encryption technology would render a crypto asset more secure relative to other crypto assets at a point in time. In other words, security is a cross-sectional attribute of a crypto asset.

Stability of a crypto asset reflects its vulnerability to internal attacks that take advantage of the *structure, governance and network architecture* of its blockchain. A decentralized structure, consensus governance and peer-to-peer network architecture of a blockchain are designed to ensure that there is no single point of failure. However, solving the single-point-of-failure problem for a digital system comes with a significant increase in its attack surface [40]. Vulnerabilities in the blockchain structure and governance include the risk of forks, possibility of orphaned blocks and exploitation of a simple majority consensus protocol (the so-called 51 per cent attack). Vulnerabilities in the network architecture include domain name system attacks, border gateway protocol attacks and distributed denial of service attacks among others. Stability features of an asset make its owner more or less vulnerable to a possibility of being digitally defrauded. Improvements in the structure, governance and network architecture can be achieved by adopting more credible legal, regulatory, and self-regulatory (e.g. consensus protocol) attributes. For example, stability of a crypto asset could also be improved if it can be credibly represented as an off-ledger liability on an identifiable entity such as a central bank, a foundation, a limited liability legal entity or a special purpose vehicle among others. Thus, stability is a time series attribute of a crypto asset that reflects its ability to retain value across time for a given level of security.

### 2.1.1. Classification of crypto assets

We consider security and stability as fixed exogenous attributes that take values in the intervals $s \in [0, 1]$ and $\xi \in [0, 1]$, respectively. Without loss of generality, we assume that crypto assets with initial characteristics $(s, \xi)$ can be obtained from the app for free, because the code for their generation is available as 'open source'.

Using this taxonomy, we can parametrize different classes of crypto assets. For expositional purposes, we will focus on four macro assets classes—high security/high stability, low security/high stability, high security/low stability, and low security/low stability—which we can identify with existing types of crypto assets. In particular, we can label central bank digital currencies (CBDCs) as high security/high stability assets, stablecoins as low security/high stability, cryptocurrencies as high security/low stability, and finally crypto tokens as low security/low stability. In the following, we will briefly introduce the crypto assets and provide an explanation of their categorization.

A CBDC can be defined as either a digitally native form of fiat currency of a country or a balance held in a digital form in a reserve account at the country's monetary authority. If issued by a credible monetary authority, it could be deemed very stable, as it would represent an outright (off-ledger) liability of the monetary authority. It can also be made very secure by using a combination of advanced cryptographic technologies, encryption algorithms and cyber defence capabilities administered by the centralized monetary authority [41–44].

*Stablecoins* are crypto assets whose values are pegged to baskets of fiat currencies or cash equivalents, existing financial instruments, physical assets such as commodities, as well as baskets of other crypto

**Table 1.** Crypto assets classification according to their *security* ($s \in [0, 1]$) and stability ($\xi \in [0, 1]$) parameters. We also divide the stability and security range in sub-intervals [$s$, $\xi < 0.5$] and [$s$, $\xi \geq 0.5$] to categorize macro classes of crypto assets (see §1 for details on the different assets classes) for expositional purposes according to our framework.

increasing stability →

| *security/ stability* | high stability $\xi \geq 0.5$ | low stability $\xi < 0.5$ |
|---|---|---|
| **high stability** $s \geq 0.5$ | central bank digital currencies | cryptocurrencies |
| **low stability** $s < 0.5$ | stablecoins | crypto tokens |

↑ increasing security

assets [45,46]. There exist three main stablecoins categories depending on the collateralization method. Asset-collateralized stablecoins are backed by (off-ledger) assets, e.g. fiat in USD or EUR, and are the most centralized, as they rely on a central authority serving as custodian of the assets used to back the crypto asset. Crypto-collateralized assets are more decentralized: the collateralization is done on-chain, i.e. locking digital assets on a distributed ledger platform using smart contracts. Non-collateralized stablecoins are algorithmically backed assets, where their increase or decrease of coin supply in the system is mathematically determined. The asset-collateralization mechanism renders this type of assets—similarly to the CBDCs—stable, as per our definition. Security-wise, as those assets are generally more centralized than cryptocurrencies, they may employ less sophisticated cryptographic primitives, hence they can be placed in the low-security category.

*Cryptocurrencies* are decentralized crypto assets relying on cryptography to secure the transfer of value between peers in the network [1]. The pioneering cryptocurrency, Bitcoin [4], appeared in 2008 and was followed soon after by a remarkable number of other coins, presenting very heterogeneous characteristics and improving on a subset of functionalities of Bitcoin. Notable examples are the Ethereum platform [47], where smart contracts functionalities have been introduced, or Monero and ZCash platforms [48,49] with improved user's anonymization techniques and with new implemented cryptographic primitives enhancing the capabilities of the systems. Most cryptocurrencies employ sophisticated encryption and digital signatures tools, hence they can be considered high security assets, while their decentralized nature by definition makes them less stable as a category.

*Crypto tokens* are tradable crypto assets and utilities built on distributed ledger platforms. Crypto tokens can be considered utility tokens if they grant holders access to an existing current or future product or service built on an existing distributed ledger platform, such as Ethereum. In some instances, tokens may be considered future sales, investment and participation schemes created to fund projects and may present features typical of securities, i.e. claims on future cash flows. Indeed, tokens that do fall outside this classification also exist and may offer the most diverse rights and functionalities for the users [50]. Crypto tokens normally have the same decentralization and network features of major cryptocurrencies, i.e. similar stability features (low stability). In terms of security, crypto tokens, instead, have suffered in recent years a number of attacks [51]: they are issued by smaller communities, who do not always disclose information about the cryptographic tools deployed, and are held in less secure wallets.

A summary of the crypto assets classification is shown in table 1. This simplified binary classification framework yields qualitatively the same results compared to the scenario where security and stability are assumed to be continuous. The continuous modelling allows for accommodating within-class heterogeneity. For instance, if we consider the stablecoins case, depending on the collateralization method they are based on (see §1), they may be classified with very heterogeneous 'stability'

and 'security' parameter values, despite belonging to the same macro crypto asset class (low security $[s < 0.5]$/high stability $[\xi \geq 0.5]$).

We, of course, recognize the limitations of representing a multitude of existing crypto assets in such a parsimonious way. Thus, our description is mainly for modelling rather than legal or regulatory purposes. It is also not intended as investment advice.

## 2.2. Demand for crypto assets

We assume that demand for the $N \gg 1$ crypto assets is driven by the probabilities of their adoption (i.e. investment in). In a real setting, the 'adoption process' would correspond, for instance, to a decision to exchange Bitcoins for Ethereum tokens. Other examples of adoption for digitally native assets that are neither tokens/coins nor serve as a currency, could include a scenario when a user may decide to use/invest less in software $A$ compared to software $B$.

Without loss of generality, we assume that at each time $t$, each of $K(t)$ investors chooses from a pair of crypto assets, $i$ and $j$ selected from the total pool. At that time, asset $i$ has the probability of adoption denoted by $a_i(t)$ and an expected return denoted by $r_i(t)$, where $r_i(t) = \sum_{m=1}^{n} x_m(t) p_m(t)$ is a weighted sum over the $n$ possible pay-offs or returns $x_m(t)$ by the probability that the given pay-off is achieved.

Under these assumptions, the adoption process for crypto asset $i$ at time $t$ is a Bernoulli process with probability

$$f(\ell, a_i(t)) = a_i^\ell(t)(1 - a_i(t))^{1-\ell}, \tag{2.1}$$

where $\ell = \{0, 1\}$ is a binary indicator that asset $i$ has not been ($\ell = 0$) or has been ($\ell = 1$) adopted and $a_i(t)$ is the probability of adoption for asset $i$ at time $t$. The probabilities of adoption for assets $i$ and $j$, $i, j = 1, \ldots, N$ are updated in time as follows:

$$\begin{aligned} a_i(t+1) &= \max(a_i(t) - \delta_i(t), 0) \\ a_j(t+1) &= \min(a_j(t) + \delta_j(t), 1). \end{aligned} \tag{2.2}$$

In general, the increment $\delta_i(t)$ is a function ($g(\cdot)$) of the number $u_i(t)$ of adopters of asset $i$ at time $t$, namely $\delta_j(t) = g(u_j(t))$. In the context of digitally native assets, popularity indicators (i.e. number of adopters) are indeed readily available to investors. In fact, they are explicitly advertised to drive adoption. Examples include the number of downloads for a digital object (a file, an app), the number of users on a digital platform, the number of wallets on a payment platform, the number of transactions processed by a crypto exchange. In this regard, an investor deciding to exchange Bitcoins for Ethereum tokens would be heavily influenced by the number of other investors using one asset over the other as a means of payment.

As a baseline modelling scenario, we make the following assumptions. First, we set the number of investors $K(t) = K$ constant and equal to $N/2$. Second, we assume that assets $i$ and $j$ presented to each investor are chosen uniformly at random from the total pool of assets. The modelling choice of comparing assets in randomly chosen pairs can also be relaxed to include different mechanisms, but this modification does not drastically impact the analysis nor the results. For instance, within the current framework, we could easily consider the case where an investor compares asset $i$ against the 'average features' of all other assets. Third, we assume that $\delta_i(t) = \delta > 0$ is constant.

We also assume that *proposed* adjustments from all investors over all possible pairs of crypto assets are collected by the crypto app as described in the next subsection. The app then presents each investor with a recommendation to select $i$ over $j$ or not. Investors may choose the app's recommendation at face value or may make their own choices depending on their trust in the app (the trust is not explicitly modelled in this paper). After investors made their choices, the app (sequentially) calculates the new vector of probabilities of adoption for each asset. Then, the app also updates expected returns based on the new probabilities of adoption and global parameters, as detailed below.

## 2.3. The crypto app

As mentioned in §1, preferences driving investments in crypto assets may differ from the mechanisms that apply to standard financial assets (bond, stocks, etc.). Like other assets though, individual crypto assets are held in expectation of some future economic benefits or 'returns' of some sort, which are often related to the aggregate return on the 'market' for the asset class as a whole [7].

We can frame, for expositional purposes, the preference revelation process and optimal selection of crypto assets, together with the above-mentioned considerations, in the following way. We assume

that investors communicate their preferences for investing in crypto assets by interacting with a digital platform defined as 'crypto app'. The app stores information about available crypto assets, which are provided to its users wishing to make investment decisions over those assets. The app collects data about suggested preferences from all investors, calculates the overall state of the 'market' to date, and then provides investor-specific optimal recommendations in the space [$a$, $r$], i.e. for assets' probabilities of adoption and for their expected returns, respectively.

In numerous contexts, digital platforms that facilitate the consumption of (digital) assets, e.g. e-books, movies and music records, are proliferating. They typically also include a recommender function based on the revealed preferences of all of its users to date. We simply extend this logic by assuming that the crypto app serves as an optimizing recommendation tool for investors.

We assume that the app behaves honestly, i.e. the app does not manipulate the information while learning about the investors' behaviour and providing recommendations. In a real setting, to protect the privacy of the users submitting their preferences, the app may be equipped with cryptographic tools to encrypt information and calculate aggregated quantities without accessing or disclosing information submitted by the individual user.

### 2.3.1. Specifications

In a real-life application, as a standard procedure, upon signing up to the service, users would be provided with specifications describing (i) the type of data collected and offered, and (ii) how those data will be used to provide investment recommendations.

The app stores information about the $N$ available assets together with their associated features $s_i$, $\xi_i$ $\forall i = 1, \ldots, N$, as well as their position in the adoption–expected return space at each point in time. The app also computes a global parameter, the total expected return, $R_{\text{tot}}(t) \in \Gamma(t)$, defined as

$$R_{\text{tot}}(t) = \sum_{i=1}^{N} a_i(t) r_i(t), \tag{2.3}$$

where $a_i(t)$ represents the $i$-th asset adoption at time $t$ and the quantity $r_i(t)$ is its expected return. This global parameter will be used by the app to evaluate whether an investor's choice to decrease (or increase) their propensity towards adopting a given asset will induce an overall decrease or increase in total expected return.

Once an investor proposes a change in adoption, as described in equation (2.2), the app calculates the total expected return $R_{\text{tot}}(t)$ if the proposed changes were adopted. The app also calculates the difference $\Delta R_{\text{tot}}(t) = R_{\text{tot}}(t-1) - R_{\text{tot}}(t)$ in total expected return, which is used to provide a recommendation to investors on whether or not they should proceed with the suggested choice.

We assume that the probability of accepting the changes proposed in equation (2.2), $P_{i,j}(\Delta R_{\text{tot}}, \mathbf{f})$, directly depends on the difference between the total expected returns, $\Delta R_{\text{tot}}(t)$, before and after modifying the adoptions $a_i(t)$, as well as on the intrinsic features of the crypto assets being compared, $\mathbf{f}$, i.e. their stability and security parameters.

We assume that the app uses the following functional form to calculate the probability of recommending the changes in adoption between assets $i$, $j$ and thus transitioning from the old states $a_i$, $a_j$ to the new proposed ones $\tilde{a}_i$, $\tilde{a}_j$ (according to the update in equation (2.2)),

$$P_{i,j}(\Delta R_{\text{tot}}, \mathbf{f}) = P(a_i \rightarrow \tilde{a}_i, a_j \rightarrow \tilde{a}_j) = \frac{1}{(1 + e^{\Delta R_{\text{tot}}})(1 + e^{\Delta s})(1 + e^{\Delta \xi})}, \tag{2.4}$$

where $\Delta R_{\text{tot}}(t) = R_{\text{tot}}(t-1) - R_{\text{tot}}(t)$ is the difference in total returns before and after the proposed move is made, $\Delta s = s(i) - s(j)$ is the difference between the security parameter of asset $i$ and $j$, and analogously $\Delta \xi = \xi(i) - \xi(j)$ is the difference between the stability parameters of the two assets. Under this functional specification, which is reminiscent of the standard Glauber dynamics in statistical physics [52], there is a higher chance of transitioning to the new state if $\Delta R_{\text{tot}}(t) < 0$, i.e. if there is a gain in total return by changing the weights, or if the asset's adoption chances are increased by more desirable security and stability configurations, i.e. if $\Delta s < 0$, $\Delta \xi < 0$.

The app may also require a fee for providing suggestions to investors and processing calculations using proprietary data stored on their servers. A fixed transaction cost $c$ may be subtracted from the calculation of the total expected return. This cost will indeed affect $\Delta R_{\text{tot}}$ and the probability of accepting the proposed change in adoption (see equation (2.4)),

$$\Delta R'_{\text{tot}} = R_{\text{tot}}(t-1) - R_{\text{tot}}(t) - c. \tag{2.5}$$

If the costs associated with the proposed change exceed the potential increase in expected returns, the investor may be less inclined to make the change effective. Adding a transaction cost has essentially no other effects than inducing a friction term that slows down the adoption dynamics.

At each time $t$, each one of the $K$ investors proposing a change in adoption, as explained in §2.2, may decide to make this change effective or not based on the information calculated by the app, namely the probability in equation (2.4). Each investor generates a uniform random threshold $p_K \in [0, 1]$: the change will be made effective if $P_{i,j} > p_K$ and discarded otherwise.

The $K$ investors will be sequentially asked to cast their preferences (i.e. change adoption probabilities of assets $i$, $j$ or not). Once all $K$ investors have submitted their choices, the app will update and store the new vector $\mathbf{a}' = \{a'_1, a'_2, \ldots, a'_N\}$ of crypto assets adoptions.

Changes in adoption affect the expected returns of assets: at each step the app recalculates and updates the expected return for every crypto asset $i$. Here, we assume that changes in expected returns for each asset are driven by two main factors: (i) changes in the adoption rate, and (ii) intrinsic features of the asset. Specifically, we assume that the app updates the expected return of crypto asset $i$ according to the following rule:

$$r_i(t) = r_i(t-1) + \Delta a_i(t) + \eta_i(t). \tag{2.6}$$

$\Delta a_i(t) = a_i(t-1) - a_i(t)$ represents the change in adoption of asset $i$ from the previous steps of the dynamics $t-1$ to the current one $t$. The term $\eta_i(t)$ represents a random component or noise generated from a Gaussian distribution with mean $\mu = 0$ and variance $\sigma_i = f(\xi_i)$, which is a function of the stability parameter $\xi_i$ of asset $i$. In the following, we will define $f(\xi_i) = 1/\xi_i$: the higher the stability of the asset, the smaller the fluctuations in expected returns.

### 2.3.2. Optimal recommendations

The dynamics over the adoption probabilities equilibrates at a time $t^\star$, when $a_i(t^\star) = a_i(t^\star - 1) = a_i^\star$, $\forall i = 1, \ldots, N$. This corresponds to having a small chance of accepting any new proposed state in probability of adoption for all the investors over all assets pairs $i$, $j$:

$$P_{i,j} - p_K < \epsilon \; \forall i, j. \tag{2.7}$$

According to equation (2.6), even when the adoption probabilities have stabilized, i.e. at $t^\star$, the expected returns would still be randomly fluctuating, due to the random noise $\eta_i(t^\star)$:

$$r_i^\star := r_i(t^\star) = r_i(t^\star - 1) + \eta_i(t^\star). \tag{2.8}$$

The app will then need to provide optimal recommendation for the volatility of the expected returns. We assume that this recommendation is computed over the different $\kappa = 1, \ldots, 4$ classes of assets we considered in this model. The way this is done is by attempting to maximize the total expected return of each subclass while simultaneously keeping its volatility bounded by minimizing the distances of each asset from the 'centre of mass' of their respective clusters.

At $t^\star$, the app computes the mean adoption probability $\bar{a}^\star = (1/N_\kappa) \sum_{i \in \kappa} a_i^\star$ and mean return $\bar{r}^\star = (1/N_\kappa) \sum_{i \in \kappa} r_i^\star$ for each class $\kappa$, and then runs the following optimization problem per asset class to find the pair $(a_i^{\text{fin}}, r_i^{\text{fin}})$, $\forall i$:

$$\min_{a_i, r_i, i \in \kappa} \sqrt{(a_i - \bar{a}^\star)^2 + (r_i - \bar{r}^\star)^2},$$
$$\text{s.t.} \sum_{i \in \kappa} r_i a_i \geq \sum_{i \in \kappa} r_i^\star a_i^\star. \tag{2.9}$$

All these steps can be interpreted as part of an optimal selection process of crypto assets performed by the investors using information and recommendation provided by the crypto app. At the end of the selection crypto assets will naturally cluster in different regions of the adoption-return space, depending on their features and the investors' preferences.

A schematic of the crypto assets dynamics is presented in figure 1. In figure 1, we schematically describe how assets may move in the adoption–expected return space starting from the initial configuration as their values are updated. As users start interacting with the app and taking advantage of its recommendations, the local knowledge on the assets, namely their adoption probabilities $a_i(t)$ and their expected returns $r_i(t)$, as well as the global state are updated. Different colours represent assets belonging to different classes, depending on their attributes.

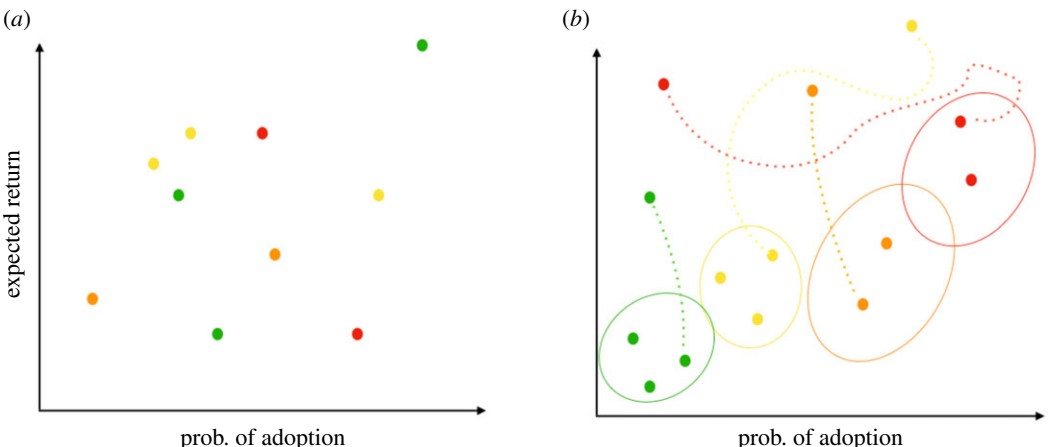

**Figure 1.** Schematic of the crypto assets dynamics. Circles of different colours represent crypto assets with different security–stability features. (*a*) At the beginning of the simulation of the dynamics, crypto assets of different types (depending on their security–stability parameters) are randomly placed in the probability of adoption–expected return space. (*b*) When investors start making their adoption choices, which will in turn impact the assets' expected returns, crypto assets start moving and clustering in different regions of the probability of adoption–expected returns space depending on their features and the investors' choices.

### 2.3.3. Settings

The app can accommodate user-specific settings. Each user can define the weights it places on assets features and global variables so that the acceptance probability in equation (2.4) can reflect investors' preferences, for example as follows:

$$P(a_i \rightarrow \tilde{a}_i, a_j \rightarrow \tilde{a}_j) = \frac{1}{(1 + e^{\beta_0 \Delta R_{tot}})(1 + e^{\beta_1 \Delta s})(1 + e^{\beta_2 \Delta \xi})}, \tag{2.10}$$

where the $\beta$s parameters represent investors' attitudes towards security, stability of individual assets as well as a global state of the system. Those parameters are used to tune (e.g. increasing or decreasing) the importance of individual assets attributes. Indeed, different investors may score the same crypto asset differently depending on their perception of its usefulness (e.g. in terms of the business model) or of the risks associated with it.

The full scheme highlighting the main components of the model, namely supply, demand and the features of the crypto app described above, is shown in figure 2. In this scheme, we show how investors would interact with the digital platform and the flow of information from and towards the app.

## 3. Results

In this section, we simulate $K$ investors interacting with $N$ crypto assets via a crypto app and we analyse possible outcomes for the crypto assets market in terms of adoption and expected returns. We consider different scenarios in terms of investors' attitudes towards assets' attributes. In §3.1, we analyse the case of homogeneous investors, all characterized by the same parameters $\beta_i$, $i = 0, 1, 2$. In the context of the crypto app setting, this means that all users decide to use parameters set up by the app itself, or that the app calculates average $\beta$s using the input of all users. In §3.2, we extend the model to consider heterogeneous investors with parameters $\beta$s extracted from different probability distributions. In the app context, each investor sets up their own level of preferences.

We initialize the dynamics by creating $N$ different crypto assets $\kappa_i$, $i = 1, \ldots, N$, where each asset $i$ is defined by a vector of intrinsic features $\mathbf{f}_i$. Features for each asset $s_i$, $\xi_i$, $i = 1, \ldots, N$ can assume a value in [0, 1], and will be randomly generated from $\pi(s)$, $\hat{\pi}(\xi)$, respectively, the probability density functions of the security and stability parameters. Features can be extracted from different distributions, yielding a different set of assets investors can buy or sell. In this way, we generate different types of assets, belonging to the four main subfamilies.

In these simulations, the assets features will not evolve in time or adapt, but will be considered fixed throughout the adoption and investment process. Moreover, the number of assets $N$ available to investors

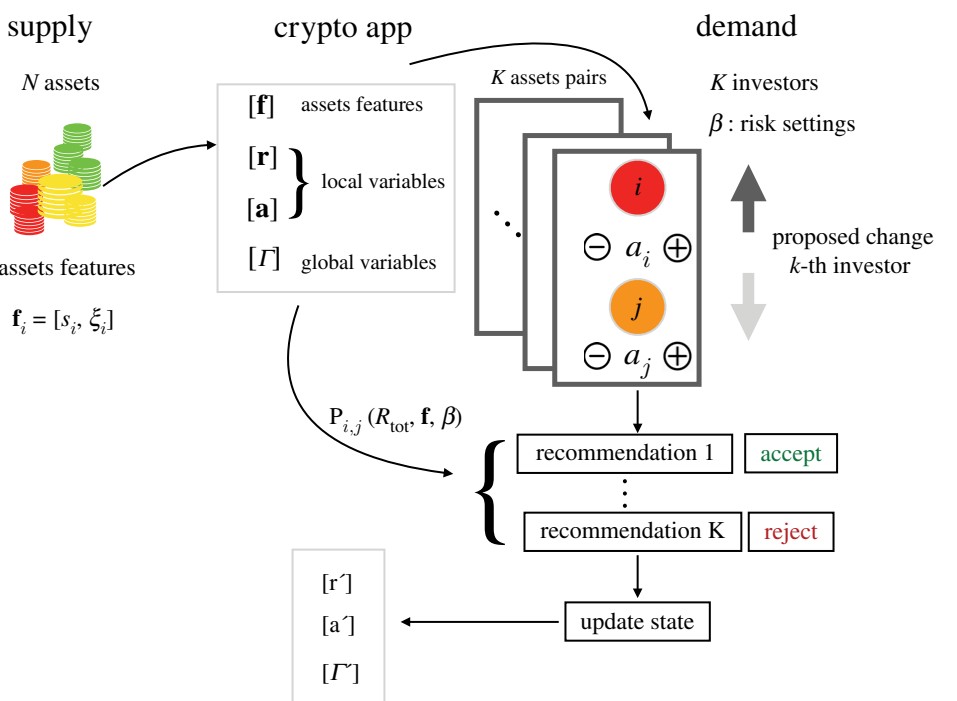

**Figure 2.** Scheme of the main components of the optimal selection model. Supply: the crypto assets landscape is composed of $N$ assets with security ($s$)–stability ($\xi$) features contained in the vector $\mathbf{f}_i$, $\forall i = 1, \ldots, N$. Crypto app: this digital platform (i) aggregates information [$\Gamma$] about the supply [$\mathbf{f}$], expected returns [$\mathbf{r}$] and investors' adoption preferences [$\mathbf{a}$] at previous times. (ii) It stores information about adoption and (iii) provides recommendations on adoption choices. Demand: $K$ investors sequentially update and provide information on their adoption preferences by comparing via the app pairs of crypto assets. The $\beta$ parameters tune the investors' attitudes towards assets' features.

will remain constant, assuming that in the timescale of observation of the market no new assets will be created nor existing ones will disappear (due to default or failure of the platform).

The dynamics is in discrete time, and simulations are run for $n_s$ steps until convergence. It is important to note that for the sake of the simulations, to guarantee convergence in the return space we will not exactly calculate the full optimization at every step as described in equation (2.9), but we will implement an effective process by rescaling by an arbitrary fixed quantity the adoption and return values across different classes at each step, until the distance minimization condition is satisfied.

Specifically, we use the following protocol. At each step $t$:

(i) we compute the 'centre of mass' for $N_\kappa$ assets belonging to the same $\kappa$ class,

$$(\bar{a}(t), \bar{r}(t)) = \left( \frac{1}{N_\kappa} \sum_{i \in \kappa} a_i(t), \frac{1}{N_\kappa} \sum_{i \in \kappa} r_i(t) \right). \tag{3.1}$$

(ii) We check whether the distance of asset $i$ from the centre of mass, $d_i = \sqrt{(a_i(t) - \bar{a}(t))^2 + (r_i(t) - \bar{r}(t))^2}$, exceeds a fixed threshold $\theta$.

(iii) If $d_i \geq \theta$, we adjust both $r_i(t)$ and $a_i(t)$ as follows:

$$r_i^{\text{adj}}(t) = r_i(t) - \frac{\bar{r}(t) - r_i(t)}{2} \tag{3.2}$$

and

$$a_i^{\text{adj}}(t) = a_i(t) - \frac{\bar{a}(t) - a_i(t)}{2}. \tag{3.3}$$

We iterate this process for all assets until convergence, when $r_i^{\text{fin}} = r_i^{\text{adj}}(t^\star)$ and $a_i^{\text{fin}} = a_i^{\text{adj}}(t^\star)$ defined in §2.3.2.

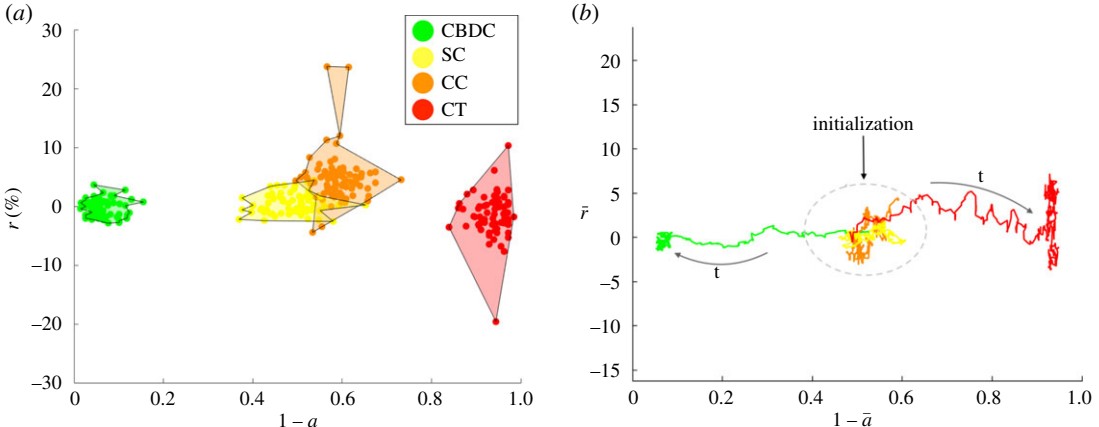

**Figure 3.** (a) Optimal assets selection outcome in the adoption–expected return space. Simulation with $N = 300$ crypto assets, $\delta = 0.1$, $\beta_0 = \beta_1 = \beta_2 = 1$ and $n_s = 400$. (b) Evolution of centre of mass trajectories in the return ($\bar{r}$) and adoption ($1 - \bar{a}$) space as a function time ($t$) for the different asset classes (see equation (3.1)). Legend—green, Central Bank Digital Currency (CBDC); orange, cryptocurrencies (CC); red, crypto tokens (CT); yellow, stablecoins (SC).

## 3.1. Homogeneous investors case

We simulate different crypto-ecosystems by assuming that the assets features—security $s$ and stability $\xi$—are both generated from uniform probability distributions $\pi(s)$, $\hat{\pi}(\xi)$ in [0, 1]. We consider a representative investor (all investors have the same parameters $\beta_{0,1,2}$) with different propensity levels ($\beta_1$, $\beta_2$) for the two types of features, security and stability, and we observe the outcome of the agent's decisions.

Let us first consider the case of a representative investor with $\beta_0 = \beta_1 = \beta_2 = 1$. In this scenario, all factors—total expected return, security and stability parameters of the assets—contribute equally to the acceptance probability in equation (2.10). The representative investor will, indeed, change the assets' adoption probabilities (according to equation (2.2)) by optimizing both with respect to the total return and the assets' parameters, security and stability. Equivalently, the change will be accepted if there is an increase in total return, and if the asset whose adoption is increased is more secure and stable than the one we are comparing it with. Under these conditions, as shown in figure 3, the assets most likely to be adopted in the future are CBDCs and stablecoins, while cryptocurrencies and crypto tokens are the least adopted but also the ones with the highest fluctuations in expected return (less stable). Indeed, given that $\beta_1 = \beta_2 = 1$, there is neither a strong aversion nor propensity from the investor's point of view towards security or stability features of the assets. In figure 3b, we monitor the convergence via the dynamics of the centre of mass (left), as defined in equation (3.1). In figure 5a, we show the evolution in time of the total expected return (figure 5b) (see equation (2.3)).

By choosing a very small parameter $\beta$, the dynamics is essentially driven by the maximization of the total return and will not be strongly affected by the intrinsic assets' features. In figure 4, we show the assets dynamics for $\beta_1 = \beta_2 = 0.01$: all assets mean adoption probabilities are centred around $a_i \sim 0.5$, $\forall i = 1, \ldots, N$ (i.e. they will have a 50% chance of surviving) indicating that the representative investor is not biased towards investing or not on a given asset $i$ (based on the asset's features). The dynamics is mostly driven by fluctuations in total return, as is also noticeable from the average behaviour of each asset class in figure 5b. By observing the total return in time $R_{\text{tot}}(t)$ in figure 5b, we clearly note, despite the stochasticity, an upward trend in time, differently from the case with $\beta > 0$, where the optimization over the assets' features as well led to a less steady increase of the total return for the representative investor (figure 5a). As shown in figure 4, the assets will differentiate mostly in terms of their expected return, with cryptocurrencies and crypto tokens experiencing the widest fluctuations (due to their lower stability parameters $\xi$s).

Note that the results are not an artefact of the choice of the acceptance probability in equation (2.4): different functional forms with similar monotonic behaviour yield qualitatively similar results. In our model, we opted for a transition probability in a form compatible with a Glauber dynamics [52]: we essentially set up a Markov chain, or equivalently a local dynamical rule specifying under which conditions a system in an initial state $\mu$ should transition to a new state $\nu$. By making a parallel with statistical physics, assets in our space are moving on a rough landscape and they are exploring it, via the Glauber dynamics, searching for global minima of their complex energy function. In some cases, a move that decreases the total returns (i.e. $\Delta R_{\text{tot}}(t) > 0$), or which increases the adoption probabilities of riskier

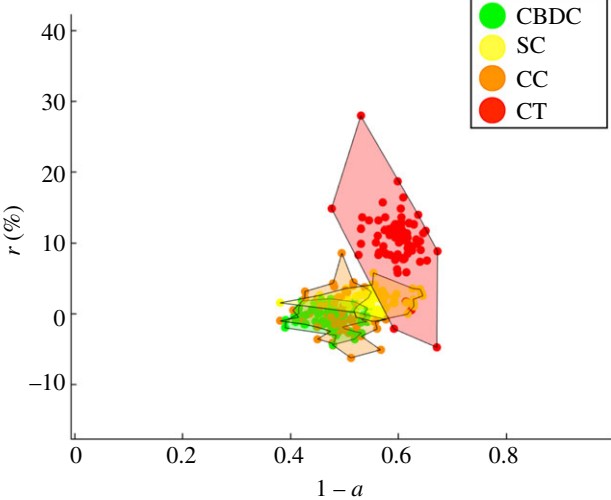

**Figure 4.** Optimal assets selection outcome in the adoption–expected return space. Simulation with $N = 300$ crypto assets, $\delta = 0.1$, $\beta_0 = 1$, $\beta_1 = \beta_2 = 0.01$ and $n_s = 300$. Legend—green, Central Bank Digital Currency (CBDC); orange, cryptocurrencies (CC); red, crypto tokens (CT); yellow, stablecoins (SC).

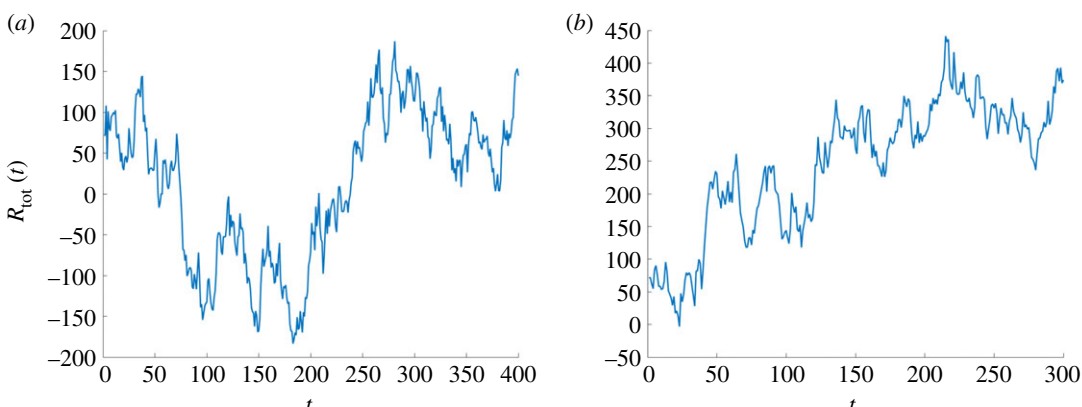

**Figure 5.** Total expected return in time for different $\beta$ parameters. (a) Simulation with $N = 300$, $\delta = 0.1$, $\beta_0 = 1$, $\beta_1 = \beta_2 = 1$ and $n_s = 400$. (b) Simulation with $N = 300$, $\delta = 0.1$, $\beta_0 = 1$, $\beta_1 = \beta_2 = 0.01$ and $n_s = 300$.

assets (e.g. $\Delta s > 0$) might still be accepted with a certain probability, according to equation (2.4): indeed, it is often necessary to move against the gradient that would push the asset towards a local minimum, to fully explore the entire landscape and discover a deeper energy valley. Consistently with the parallel with statistical physics, the $\beta$-parameters in equation (2.10) act as an equivalent of an inverse temperature [53].

The parameter $\beta$ essentially defines the investor(s)'s attitudes. In particular, by setting $\beta < 0$ we model a risk-prone investor, who will be more likely to invest on less secure and/or less stable assets. For instance, in figure 6, we analyse the case of $\beta_1 = \beta_2 = -2$, where the investor has a strong propensity towards both types of features. In this case, the stable configuration for the assets is characterized by the riskiest assets, i.e. cryptocurrencies and crypto tokens, having a low probability of not being adopted. The outcome is clearly specular to the one observed for a risk-averse investor in figure 3.

Our representative investor on the crypto app may also have different attitudes towards the different assets' features. For example, an investor may be more (or less) prone to tolerate risks associated with the assets' security or stability attributes: this would correspond to setting $\beta_1 \neq \beta_2$ in the crypto app user's preferences. Indeed, according to equation (2.10), $\beta_1$ will represent the risk-aversion parameters towards risks associated with the asset's security features, while $\beta_2$ will take into account risks connected to the asset's stability.

In figure 7, we compare two opposite cases: one where the representative investor has a high aversion towards low-stability assets (high $\beta_1$), while considerations on the security of the assets do not affect the probability that a given investor will adopt them (low $\beta_2$, close to zero) with the mirroring one, with a high $\beta_2$ and a negligible $\beta_1$. In the first case (figure 7a), the most adopted assets are indeed also the

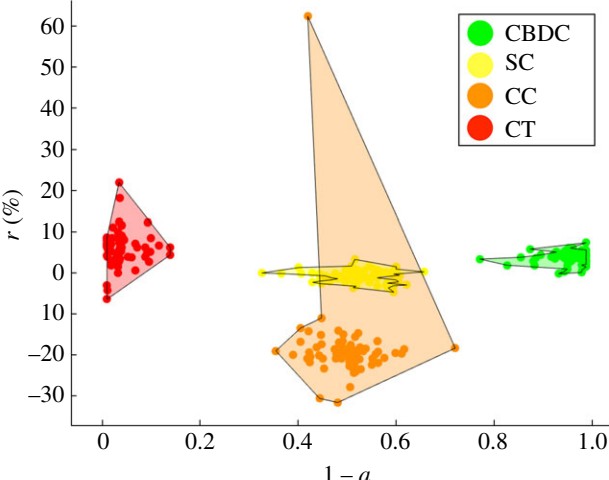

**Figure 6.** Optimal assets selection outcome in the adoption–expected return space. Simulation with $N = 300$ crypto assets, $\delta = 0.1$, $\beta_0 = 1$, $\beta_1 = \beta_2 = -2$ and $n_s = 500$. Legend—green, Central Bank Digital Currency (CBDC); orange, cryptocurrencies (CC); red, crypto tokens (CT); yellow, stablecoins (SC).

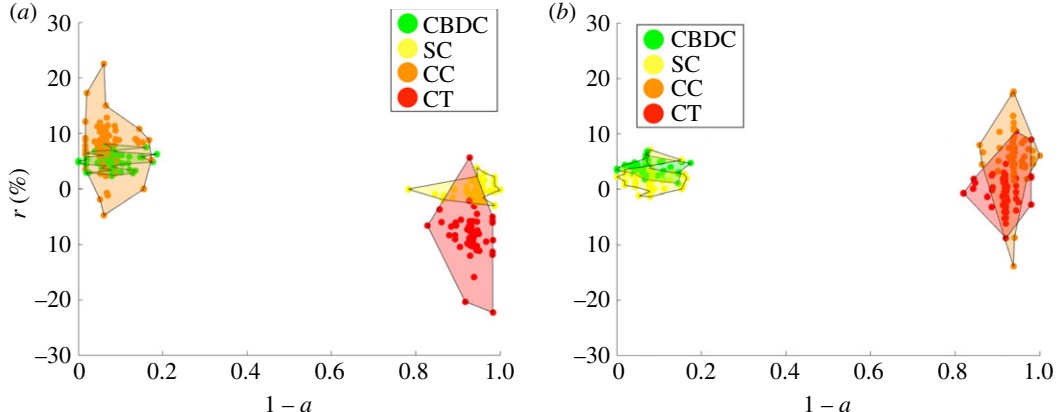

**Figure 7.** Optimal assets selection outcome in the adoption–expected return space. Simulation with $N = 300$ crypto assets, $\delta = 0.1$ and $n_s = 300$, $\beta_0 = 1$. (*a*) $\beta_1 = 2$, $\beta_2 = 0.01$. (*b*) $\beta_1 = 0.01$, $\beta_2 = 2$. Legend—green, Central Bank Digital Currency (CBDC); orange, cryptocurrencies (CC); red, crypto tokens (CT); yellow, stablecoins (SC).

most secure ones according to our classification, i.e. CBDCs and cryptocurrencies. In the second case (figure 7*b*), the representative investor has instead a predilection for more stable assets, which correspond to stablecoins and CBDCs and which now occupy the leftmost part of the diagram.

As we can conclude from these examples, the assets' dynamics displays a very rich behaviour by varying the system's parameters. In particular, the dynamics of the adoptions appears to be quite informative when predicting the chance of an asset of surviving in the future. To better monitor changes in adoption for the different assets under various conditions, we provide a normalized histogram of $1 - a$ (i.e. the mean probability of not being adopted), conditioned on the asset class (i.e. CBDCs, CTs, SCs, CCs). The results are summarized in a phase diagram-like representation as a function of the parameters $\beta_1$, $\beta_2$ in figure 8. By varying the $\beta$ parameters, the ecosystem moves from one phase to another, characterized by different outcomes for the probability of not being adopted per asset class.

### 3.1.1. Distribution of the assets' features and rescaling

So far, assets features were generated from uniform probability density functions $\pi(s)$, $\hat{\pi}(\xi)$, so that all subclasses were homogeneous in terms of number of assets per class. Indeed, different distributions of assets features yield a different composition of the crypto-ecosystem for the investors and this may affect the outcomes of the investment decisions. Interestingly, we can show that changes in the composition of the crypto-market can be rebalanced by modifying the investors' $\beta$ parameters in such a way that the same outcome is obtained in terms of adoption probability–expected return of the assets. Comparing

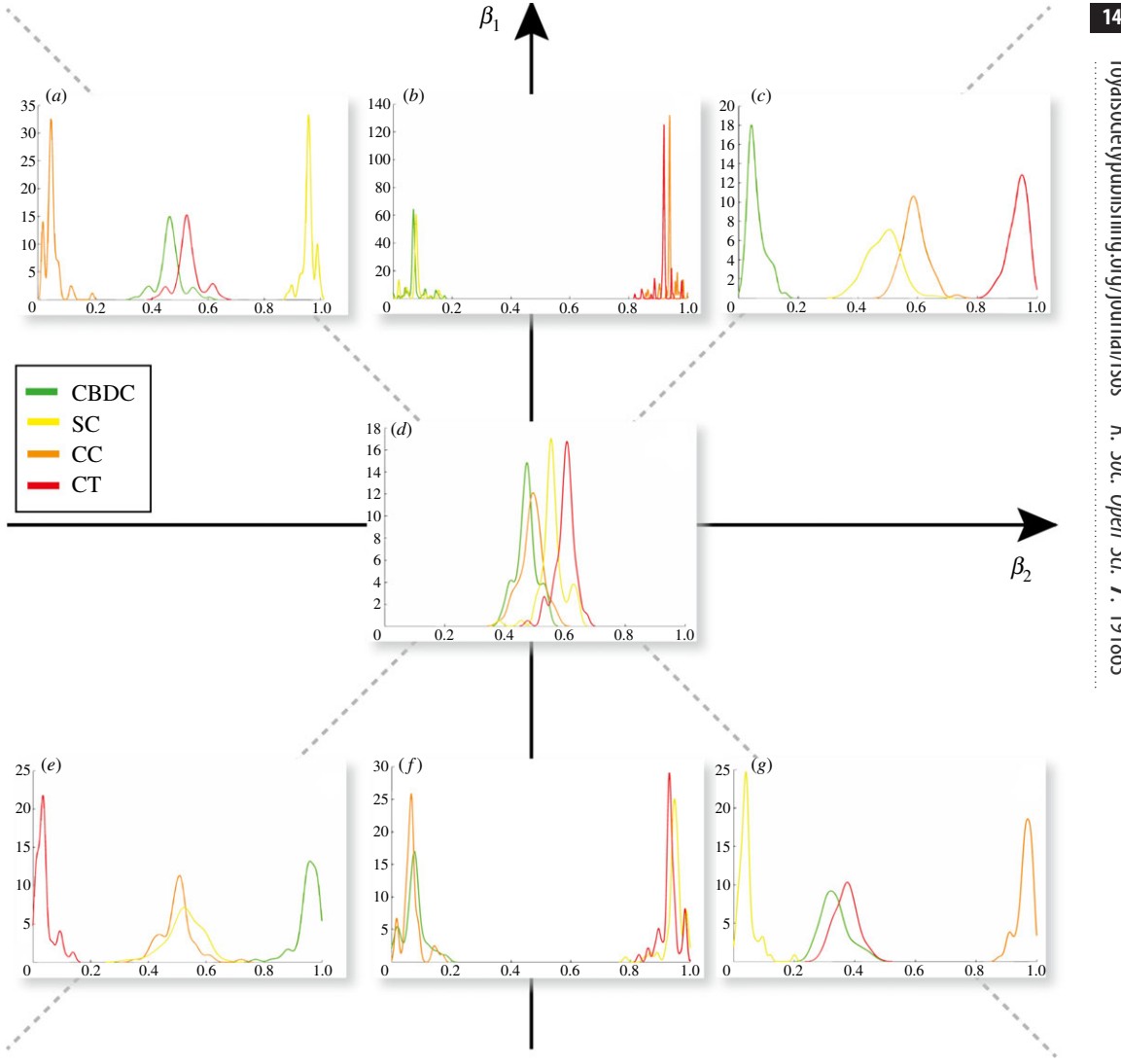

**Figure 8.** Phase diagram of the system varying the parameters $\beta_1$, $\beta_2$ with $\beta_0 = 1$. By monitoring the distribution of mean probability of not being adopted $1 - a$, conditioned on the assets' class $\kappa$, $P(1 - a|\kappa)$, we can identify different stable configurations for the system, in different regions of the parameter space. We include a few illustrative examples: (a) $\beta_1 = -2$, $\beta_2 = 2$, (b) $\beta_1 = 2$, $\beta_2 = 0.01$, (c) $\beta_1 = \beta_2 = 2$, (d) $\beta_1 = \beta_2 = 0.01$, (e) $\beta_1 = \beta_2 = -2$, (f) $\beta_1 = -2$, $\beta_2 = 0.01$, (g) $\beta_1 = 2$, $\beta_2 = -2$. In all simulations, we consider $\delta = 0.1$, $n_s = 300$, $N = 300$ and $\beta_0 = 1$.

two ecosystems where features were generated from $\pi(s)$, $\hat{\pi}(\xi)$ and $\pi'(s)$, $\hat{\pi}'(\xi)$, one sees that the acceptance probability in equation (2.10), hence the possible outcomes for the adoption probabilities of the different asset classes may change. For simplicity, we consider $\beta_1 = \beta_2$ in equation (2.10) and we ignore the returns distribution in the acceptance probability for the following estimate: $P \propto 1/(1 + e^{\beta\Delta s})(1 + e^{\beta\Delta\xi})$. Given $\beta$ in the first ecosystem, what would be $\beta'$ for the second ecosystem, yielding the same acceptance probabilities, i.e. $P = P'$? We can produce a rough estimate of $\beta'$ by imposing

$$\left\langle ((1 + e^{\beta\Delta s})(1 + e^{\beta\Delta\xi}) \right\rangle_{\pi(s),\hat{\pi}(\xi)} \overset{?}{=} \left\langle (1 + e^{\beta'\Delta s})(1 + e^{\beta'\Delta\xi}) \right\rangle_{\pi'(s),\hat{\pi}'(\xi)'} \tag{3.4}$$

where we average over the probability density functions (pdfs) of the two features in the two ecosystems. The average can be explicitly written as follows:

$$\langle (1 + e^{\beta\Delta s})(1 + e^{\beta\Delta\xi}) \rangle_{\pi(s),\hat{\pi}(\xi)} \simeq \sum_{i,j} \frac{n_i}{N} \frac{n_i - \delta_{i,j}}{N} \int_{S(i)} \mathrm{d}\pi(s) \int_{\Xi(i)} \mathrm{d}\hat{\pi}(\xi) \int_{T(j)} \mathrm{d}\pi(\tau)$$

$$\times \int_{R(j)} \mathrm{d}\hat{\pi}(\rho)(1 + e^{\beta(s-\tau)})(1 + e^{\beta(\xi-\rho)}), \tag{3.5}$$

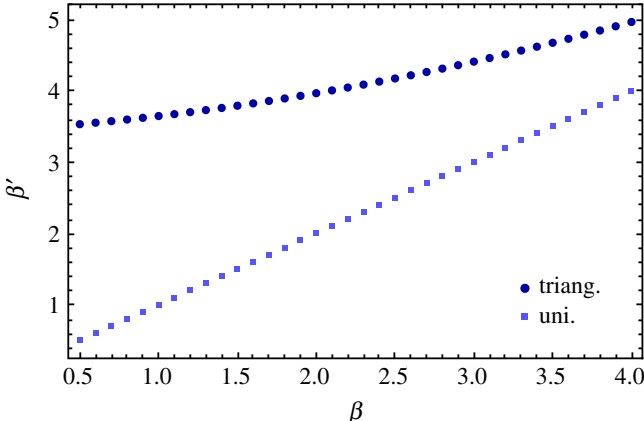

**Figure 9.** Numerical estimation of $\beta'$ as a function of $\beta$ according to equation (3.4), given $\pi'(s)$, $\hat{\pi}'(\xi)$ uniform and (i) $\pi(s) = 1$, $\hat{\pi}(\xi) = 1$ (uniform pdf with $s$, $\xi \in [0, 1]$) and (ii) $\pi(s) = 2s$, $\hat{\pi}(\xi) = 2\xi$ (triangular pdf with $s$, $\xi \in [0, 1]$).

where $n_i = N \int_{S(i)} ds\, \pi(s) \int_{\Xi(j)} d\xi\, \hat{\pi}(\xi)$ is the average number of crypto assets in class $i = 1, \ldots, 4$ and $n_i/N$ is the probability of extracting an asset of class $i$. We also introduced the following notation where $ds\pi(s) = d\pi(s)$. We integrate over four regions $S(i)$, $\Xi(i)$, $T(j)$, $R(j)$ where the stability and security parameters of the assets $i$ or $j$ are defined. For instance, extracting asset $i$ from the CBDC class would mean—as defined in table 1— that the security and stability parameters lie, respectively, in the regions $s \in [0.5, 1] = S(i)$ and $\xi \in [0.5, 1] = \Xi(i)$. In equation (3.5), we are approximating the average over the pdfs by summing over all possible combinations of assets $j$, $k$ extracted from the different classes, weighted by the probabilities that the two selected assets belong to the a given class $\frac{n_i}{N} \frac{n_i - \delta_{i,j}}{N}$.

By solving it numerically, fixing $\beta$ we can estimate the values of $\beta'$ that would yield the same acceptance probability, hence a similar outcome for the simulations. In figure 9, we show the results of the numerical estimation of equation (3.4). We consider the trivial case where in both ecosystems the assets' features are extracted from uniform distributions, where we recover $\beta = \beta'$ and the case where we choose $\pi(s) = 1$, $\hat{\pi}(\xi) = 1$ in one ecosystem and $\pi'(s) = 2s$, $\hat{\pi}'(\xi) = 2\xi$ (triangular). This second scenario will be used in our illustrative example below and the estimated values for $\beta'$ will be fed into the simulations to observe the behaviour of the different ecosystems. Therefore, we simulate a system with $\pi(s) = \hat{\pi}(\xi) = 1$ uniform between $[0, 1]$ and one where the features are extracted from a triangular pdf in $[0, 1]$: $\pi(s) = 2s$, $\hat{\pi}(\xi) = 2\xi$. In figure 10, we summarize the result by plotting mean (and variance) of the probability of not being adopted per asset class. In general, simulations with uniform and triangular distribution yield, for the same $\beta = 1$, quite different outcomes. Nonetheless, by using the numerical estimation in figure 9, we find $\beta'$ such that the outcome of the ecosystem with uniform features would become equivalent to the one generated considering a triangular pdf for the features instead.

Changes in the composition of the crypto-ecosystem, which may yield new investments and adoption outcomes, can be offset by a correct investors' assessment of the current situation and re-estimation of their attitudes towards assets' characteristics.

## 3.2. Heterogeneous investors case

In this section, we consider a set of $K$ heterogeneous investors with different $\beta$ parameters. At each round of the dynamics, each investor picks at random a pair of assets $i, j$ and decides whether to update the adoption probabilities (see equation (2.2)) according to their own attitudes towards assets' features. Hence, we introduce two vectors $\vec{\beta}_{1,2} = (\beta_{1,2}^{(1)}, \ldots, \beta_{1,2}^{(K)})$, containing the values of the investor-specific $\beta$ parameters $\beta_{1,2}^{(k)}$ for each investor $k = 1, \ldots, K$ using the crypto app. In the simulations, the parameters can be extracted from a probability distribution depending on the heterogeneity of the investors population that we aim to obtain. We will indicate the probability distributions, which $\vec{\beta}_{1,2}$ are extracted from, as $\hat{\Phi}_{1,2}$.

In this case, the acceptance probability in equation (2.10) can be rewritten as follows:

$$P^{(k)}(a_i \to \tilde{a}_i, a_j \to \tilde{a}_j) = \frac{1}{(1 + e^{\Delta R_{tot}})(1 + e^{\beta_1^{(k)} \Delta s})(1 + e^{\beta_2^{(k)} \Delta \xi})}, \quad k = 1, \ldots, K, \tag{3.6}$$

and it will depend on the investor's idiosyncratic preferences embedded in the parameters $\beta_{1,2}^{(k)}$, $k = 1, \ldots, K$. Note that in this case, the acceptance probability (equation (3.6)) changes from

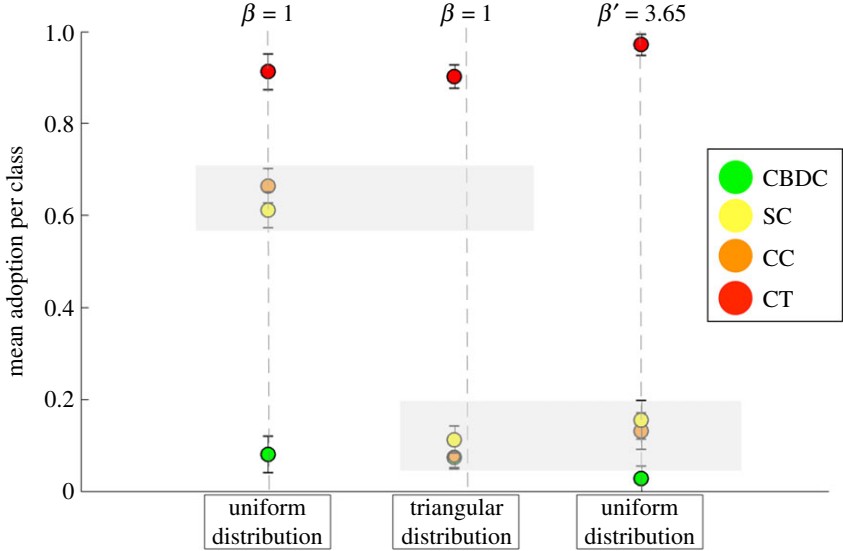

**Figure 10.** Mean (and variance) of the probability of not being adopted per asset class for the following scenario with $N = 200$ crypto assets, $\delta = 0.1$ and (i) $\beta = 1$, $\pi(s) = 1$, $\hat{\pi}(\xi) = 1$ (uniform pdf), (ii) $\beta = 1$, $\pi(s) = 2$ s, $\hat{\pi}(\xi) = 2\xi$ (triangular pdf) and (iii) $\beta' = 3.65$, $\pi(s) = 1$, $\hat{\pi}(\xi) = 1$ (uniform pdf rescaled). Legend—green, Central Bank Digital Currency (CBDC); orange, cryptocurrencies (CC); red, crypto tokens (CT); yellow, stablecoins (SC).

investor to investor. Interestingly, we can show that the choices of the different investors cannot be aggregated and lead to different macroscopic emergent behaviours. As an illustrative example, we consider a scenario in which the values of both $\vec{\beta}_1$, $\vec{\beta}_2$ are drawn from a triangular distribution with support between $[-4, 4]$, namely of the form $\hat{\Phi}_{1,2}(x) = \frac{x+4}{32}$. To gain an understanding of the effects of heterogeneity in investors' attitudes, we compare this scenario with the one obtained by considering only one representative investor with $\tilde{\beta}_{1,2} = \langle \beta_{1,2} \rangle = (1/K)\sum_{k=1}^{K} \beta_{1,2}^{(k)}$.

Results from the two simulations are shown and compared in figure 11. Indeed, the behaviour of the representative investors appear to be substantially different from the one observed as an outcome of the investment strategies played by misaligned investors. While in the homogeneous example, the representative investor is assumed to be risk averse with respect to both types of threats (i.e. related to both security and stability) as $\tilde{\beta}_{1,2} > 0$, in the heterogeneous case, the investors sets are composed of both risk-prone and risk-averse agents. Due to the randomness in the process, the effects do not average out trivially, and the case of heterogeneous investors yield potentially unanticipated outcomes. For example, in the heterogeneous case, less stable or secure assets, such as crypto tokens, have a higher chance of being adopted (i.e. $a \sim 0.8$) compared to the homogeneous case (with $a \sim 1$) because of the presence of more risk-prone agents involved in the process (figure 11).

In summary, by considering that all investors are characterized by the same parameters $\beta_1$, $\beta_2$, we are assuming that agents in the market have aligned strategies and similar propensity towards risk. In this 'homogeneous investors' case, the behaviour of multiple agents can be, therefore, described by a representative investor with parameters $\beta_1$, $\beta_2$. Introducing misaligned investors with opposite strategies determines a non-trivial behaviour in the system together with the emergence of new stable configurations in the market. Indeed, disseminating contrasting views, which may affect investors' opinions on the assets and the associated risks, may destabilize the ecosystem yielding unexpected scenarios for the investments.

## 4. Discussion

We offer a framework to classify crypto assets into major stable subclasses in terms of two main intrinsic asset's features, namely the asset's security and stability. By considering the behaviour of different types of investors, driven by their attitudes towards assets' attributes ($\beta$ parameters), we explore different outcomes for the investments in the crypto-ecosystem and the future—in terms of their adoption probability, hence their chances of surviving—of the crypto assets.

For the time being, we fixed the number of assets, the number of investors, the assets' features and investor preferences throughout the analysis. In principle, any of these features could dynamically evolve together with the asset's adoption probabilities and expected returns. For instance, an asset initially

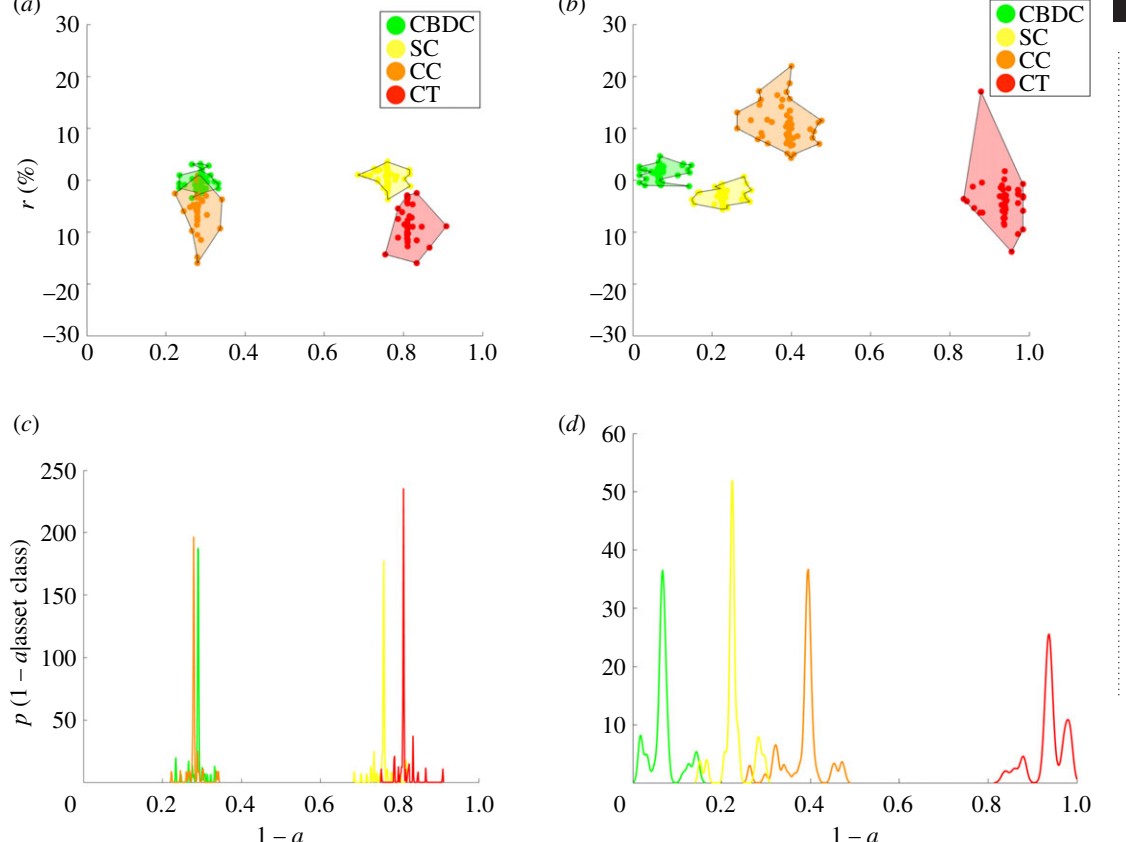

**Figure 11.** Simulation with $N = 200$ crypto assets, $\delta = 0.1$ and $n_s = 300$. (*a,b*) Optimal assets selection outcome in the adoption–expected return space for the case of heterogeneous investors (*a*) and homogeneous investors (*b*) with $\langle \beta_1 \rangle = 1.234$ and $\langle \beta_2 \rangle = 1.549$. (*c,d*) Distribution of mean probability of not being adopted $(1 - a)$, conditioned on the assets' class $\chi$, $P(1 - a|\chi)$ for the heterogeneous (*c*) and homogeneous (*d*) investor case, respectively. Legend—green, Central Bank Digital Currency (CBDC); orange, cryptocurrencies (CC); red, crypto tokens (CT); yellow, stablecoins (SC).

characterized by low security but experiencing an increase in adoption in time, may gradually try to improve its intrinsic features (e.g. investing in improving the underlying technology) to attract even more interest. Also, the number of assets may vary in time and this feature may be introduced in a further extension to the model. Assets with features that would fit the most investments trends and market demand will survive the analogue of a natural selection process for the crypto ecosystem.

Our work paves the way for further investigations whose scope is to understand the future developments of the fast-growing crypto asset ecosystem. This initial exploration lays the foundation for designing successful investment strategies and predicting adoption scenarios under different conditions.

Data accessibility. The code used for the simulations of the theoretical model of crypto assets selection is publicly available and can be found in the Dryad Digital Repository at https://doi.org/10.5061/dryad.qfttdz0cb [54].
Authors' contributions. S.B. and A.K. conceived the study, S.B. conducted the numerical simulations, S.B. and A.K. analysed the results. All authors wrote the manuscript.
Competing interests. We declare we have no competing interests.
Funding. We received no funding for this study.

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
