## [Reviewer comments · Royal Society Open Science]

Review History

RSOS-191863.R0 (Original submission)

Review form: Reviewer 1

Is the manuscript scientifically sound in its present form?

No

Are the interpretations and conclusions justified by the results?

No

Is the language acceptable?

Yes

Do you have any ethical concerns with this paper?

No

Have you any concerns about statistical analyses in this paper?

No

Recommendation?

Reject

Comments to the Author(s)

The authors introduce an agent based model to model interactions between cryptocurrency investors. The idea of designing an agent based model where investors take decisions based on the security, stability and expected return on investments seems reasonable, although the authors do not provide any evidence that this is the case. The paper suffer from major flaws, including:

Lack of clarity about the scope of the article.

My confusion started from the third sentence of the abstract, which reads: 'Investors make choices over, crypto assets similarly to how they make choices by using a recommender app'. What does this mean?

By reading the abstract, I thought the authors had ran an experiment using an app. Instead, this work is purely numerical, but the contribution is unclear. The app designed by the authors does not exist, investors do not have global knowledge around prices and other investors choices, and they do not take decisions simultaneously. It is unlikely to imagine a scenario where that would be the case, so I am puzzled about the usefulness of these numerical analyses, which seems to be a theoretical exercise. It is important to add that although much data about cryptocurrency prices and trading are publicly available, this article never refers to or make use of actual data.

Lack of clarity and realism around the model.

I have found it hard to follow the text describing the model, which makes it even harder to understand the potential contribution of this article. What does adoption mean? Investments are not binary choices. Do I understand correctly that the decision of a single agent is responsible for the change in adoption concerning all users? This seems unreasonable. Why would the return depend only on the adoption probability?

Review form: Reviewer 2

Is the manuscript scientifically sound in its present form?

Yes

Are the interpretations and conclusions justified by the results?

Yes

Is the language acceptable?

Yes

Do you have any ethical concerns with this paper?

No

Have you any concerns about statistical analyses in this paper?

No

Recommendation?

Accept with minor revision (please list in comments)

Comments to the Author(s)

1. Using security and stability as the only two dimensions of a crypto assets simplifies the modeling, but needs to be better motivated to make the model more realistic and applicable. Importantly, the authors omit the fact that many crypto assets, especially those categorized as "crypto tokens" in the paper, are designed to have specific utilities targeting different user cohorts. That is to say, depending on their view on the "business model" of a crypto token,

different users can score the same crypto token differently, even if they value “security (technology)” the same way.

2. Page 3 lines 32-38: the authors categorize “crypto tokens” as having “low security”, but did not demonstrate why that’s the case in the description of crypto tokens.

3. While it is nice and helpful to describe four categories of crypto assets, the description in its current is way too long and not so essential for the model.

4. It is slightly confusing to categorize crypto assets with two binary features --- high or low security, and high or low stability --- and later in the model, those two features appear to be continuous.

5. Page 4, line 23-31: the description of sentiment analysis appears too verbose and strays off the central theme.

6. The placement of Figure 1 is currently too early --- many notations and variables are not explained in the preceding text. I recommend to also explain all the notations either directly in the figure, or in figure notes.

More elaborate figure notes will be appreciated for other figures as well.

7. There are numerous instances of notation clashes in the paper that hampers the understanding.

For example, “k” in Eq. 2.1. stands for whether a token is adopted or not, while on page 15 line 26 it stands for an asset, and on page 16 line 34 it stands for an investor.

Another example, “j” in Eq. 2.2 is one of the assets in an asset pair, while on page 6 line 13, “j” is a counter.

Review form: Reviewer 3

Is the manuscript scientifically sound in its present form?

Yes

Are the interpretations and conclusions justified by the results?

Yes

Is the language acceptable?

Yes

Do you have any ethical concerns with this paper?

No

Have you any concerns about statistical analyses in this paper?

No

Recommendation?

Accept with minor revision (please list in comments)

Comments to the Author(s)

The article is quite clearly written and the results are statistically well supported.

I have only some objections to initial assumptions.

The conception behind cryptocurrencies was to avoid any central banks regulations and to "give control of the money" to the people. So the CBDCs may not be seen as stable from the perspective of traditional cryptocurrency community, because central banks can easily change the supply of the currency. In the case of Bitcoin and many others cryptoassets it is fixed at some maximal level.

So in my opinion the authors should more precisely describe selected stability criteria and emphasized the fact that they may be not commonly accepted.

Minor remarks:

The variables in Fig 3: Xcm and Ycm are not explicitly defined.

line 56 page 13 space missing "assets(high..."

ref 32 was already published - see <https://www.mdpi.com/1999-5903/11/7/154>

Decision letter (RSOS-191863.R0)

16-Mar-2020

Dear Dr Bartolucci,

The editors assigned to your paper ("A Model of the Optimal Selection of Crypto Assets") have now received comments from reviewers. We would like you to revise your paper in accordance with the referee and Associate Editor suggestions which can be found below (not including confidential reports to the Editor). Please note this decision does not guarantee eventual acceptance.

Please submit a copy of your revised paper before 08-Apr-2020. Please note that the revision deadline will expire at 00.00am on this date. If we do not hear from you within this time then it will be assumed that the paper has been withdrawn. In exceptional circumstances, extensions may be possible if agreed with the Editorial Office in advance. We do not allow multiple rounds of revision so we urge you to make every effort to fully address all of the comments at this stage. If deemed necessary by the Editors, your manuscript will be sent back to one or more of the original reviewers for assessment. If the original reviewers are not available, we may invite new reviewers.

- Data accessibility

If you wish to submit your supporting data or code to Dryad (<http://datadryad.org/>), or modify your current submission to dryad, please use the following link:
<http://datadryad.org/submit?journalID=RSOS&manu=RSOS-191863>

- Competing interests

- Authors' contributions

- Acknowledgements

- Funding statement

on behalf of Marta Kwiatkowska (Subject Editor)

Associate Editor's comments:

Comments to the Author:

The reviewers have supplied extensive commentary on your paper. Given the concerns raised by reviewer 1 and reviewer 2 in particular, we would like you to revise your manuscript to address these questions and comments. Please bear in mind that the journal does not generally permit multiple rounds of revision, and if the reviewers continue to express concerns with your paper, we will not be able to accept it for publication. Good luck and we'll look forward to receiving your revision in due course.

Comments to Author:

Reviewers' Comments to Author:

Reviewer: 1

Comments to the Author(s)

The authors introduce an agent based model to model interactions between cryptocurrency investors. The idea of designing an agent based model where investors take decisions based on the security, stability and expected return on investments seems reasonable, although the authors do not provide any evidence that this is the case. The paper suffer from major flaws, including:

Lack of clarity about the scope of the article.

My confusion started from the third sentence of the abstract, which reads: 'Investors make choices over, crypto assets similarly to how they make choices by using a recommender app'. What does this mean?

By reading the abstract, I thought the authors had ran an experiment using an app. Instead, this work is purely numerical, but the contribution is unclear. The app designed by the authors does not exist, investors do not have global knowledge around prices and other investors choices, and they do not take decisions simultaneously. It is unlikely to imagine a scenario where that would be the case, so I am puzzled about the usefulness of these numerical analyses, which seems to be a theoretical exercise. It is important to add that although much data about cryptocurrency prices and trading are publicly available, this article never refers to or make use of actual data.

Lack of clarity and realism around the model.

I have found it hard to follow the text describing the model, which makes it even harder to understand the potential contribution of this article. What does adoption mean? Investments are not binary choices. Do I understand correctly that the decision of a single agent is responsible for the change in adoption concerning all users? This seems unreasonable. Why would the return depend only on the adoption probability?

Reviewer: 2

Comments to the Author(s)

- Using security and stability as the only two dimensions of a crypto assets simplifies the modeling, but needs to be better motivated to make the model more realistic and applicable. Importantly, the authors omit the fact that many crypto assets, especially those categorized as "crypto tokens" in the paper, are designed to have specific utilities targeting different user cohorts. That is to say, depending on their view on the "business model" of a crypto token, different users can score the same crypto token differently, even if they value "security (technology)" the same way.
- Page 3 lines 32-38: the authors categorize "crypto tokens" as having "low security", but did not demonstrate why that's the case in the description of crypto tokens.

3. While it is nice and helpful to describe four categories of crypto assets, the description in its current is way too long and not so essential for the model.
4. It is slightly confusing to categorize crypto assets with two binary features --- high or low security, and high or low stability --- and later in the model, those two features appear to be continuous.
5. Page 4, line 23-31: the description of sentiment analysis appears too verbose and strays off the central theme.
6. The placement of Figure 1 is currently too early --- many notations and variables are not explained in the preceding text. I recommend to also explain all the notations either directly in the figure, or in figure notes.
More elaborate figure notes will be appreciated for other figures as well.
7. There are numerous instances of notation clashes in the paper that hampers the understanding.
For example, "k" in Eq. 2.1. stands for whether a token is adopted or not, while on page 15 line 26 it stands for an asset, and on page 16 line 34 it stands for an investor.
Another example, "j" in Eq. 2.2 is one of the assets in an asset pair, while on page 6 line 13, "j" is a counter.

Reviewer: 3

Comments to the Author(s)

The article is quite clearly written and the results are statistically well supported.

I have only some objections to initial assumptions.

The conception behind cryptocurrencies was to avoid any central banks regulations and to "give control of the money" to the people. So the CBDCs may not be seen as stable from the perspective of traditional cryptocurrency community, because central banks can easily change the supply of the currency. In the case of Bitcoin and many others cryptoassets it is fixed at some maximal level.

So in my opinion the authors should more precisely describe selected stability criteria and emphasized the fact that they may be not commonly accepted.

Minor remarks:

The variables in Fig 3: Xcm and Ycm are not explicitly defined.

line 56 page 13 space missing "assets(high..."

ref 32 was already published - see <https://www.mdpi.com/1999-5903/11/7/>

Author's Response to Decision Letter for (RSOS-191863.R0)

See Appendix A.

RSOS-191863.R1 (Revision)

Review form: Reviewer 1

Is the manuscript scientifically sound in its present form?

Yes

Are the interpretations and conclusions justified by the results?

Yes

Is the language acceptable?

Yes

Do you have any ethical concerns with this paper?

No

Have you any concerns about statistical analyses in this paper?

No

Recommendation?

Accept as is

Comments to the Author(s)

The authors have extensively reviewed the manuscript. I judge the scope of the article is much clearer and the paper is ready for publication at this stage.

Review form: Reviewer 2

Is the manuscript scientifically sound in its present form?

Yes

Are the interpretations and conclusions justified by the results?

Yes

Is the language acceptable?

Yes

Do you have any ethical concerns with this paper?

No

Have you any concerns about statistical analyses in this paper?

No

Recommendation?

Accept with minor revision (please list in comments)

Comments to the Author(s)

All my previous comments and concerns have been sufficiently addressed in the new version. I only have a few minor comments for this revised manuscript.

1. In Abstract the authors claim to “predict the subset of the most widely adopted crypto assets and their features”. In effect, however, the authors are solely predicting characterizations (or features) of the adoptable assets, not the exact assets themselves. I suggest either (I) modifying the language in Abstract to more precisely describe the contribution; or (II) naming a few real-life examples of the adoptable crypto assets based on the authors’ prediction, to justify the contribution.

2. The authors use “distributed ledger” and “blockchain” interchangeably when the former is a broader concept than the latter. I suggest either (I) adding a brief explanation of the two terms and notifying the readers in advance of the interchangeable use of the two terms in the paper; or (II) being precise about their distinction, and modifying the sentence such as “[...] secured by

cryptographic technology and shared electronically via a distributed ledger (blockchain)" into "[...] secured by cryptographic technology and shared electronically via a distributed ledger, such as a blockchain".

3. The section Introduction has now been significantly expanded. I suggest splitting the section into two subsections for better readability: the first one on the general background of crypto investment, the second one introducing the model framework (the "app"). In addition, I believe the inclusion of Etoro screenshots from the authors' response letter may facilitate readers' comprehension of the imaginary app.

4. It would be ideal if the authors could make the display of the figures consistent. Specifically, for Figs 6 and 7, I suggest placing the legend for crypto types directly in the figure (as done with Fig 8), as opposed to in the figure caption. For Figs 10 and 11, legends are yet to be added.

Review form: Reviewer 3

Is the manuscript scientifically sound in its present form?

Yes

Are the interpretations and conclusions justified by the results?

Yes

Is the language acceptable?

Yes

Do you have any ethical concerns with this paper?

No

Have you any concerns about statistical analyses in this paper?

No

Recommendation?

Accept as is

Comments to the Author(s)

Thank you for your work

Review form: Reviewer 4

Is the manuscript scientifically sound in its present form?

Yes

Are the interpretations and conclusions justified by the results?

Yes

Is the language acceptable?

Yes

Do you have any ethical concerns with this paper?

No

Have you any concerns about statistical analyses in this paper?

No

Recommendation?

Accept as is

Comments to the Author(s)

This is a good paper that in its revised form is much stronger than in the original one. It covers a 'hot' topic providing insightful results and contributing strongly to the literature in this field. I undoubtedly recommend publication.

Decision letter (RSOS-191863.R1)

Dear Dr Bartolucci:

On behalf of the Editors, I am pleased to inform you that your Manuscript RSOS-191863.R1 entitled "A Model of the Optimal Selection of Crypto Assets" has been accepted for publication in Royal Society Open Science subject to minor revision in accordance with the referee suggestions. Please find the referees' comments at the end of this email.

The reviewers and Subject Editor have recommended publication, but also suggest some minor revisions to your manuscript. Therefore, I invite you to respond to the comments and revise your manuscript.

- Ethics statement

- Data accessibility

<http://datadryad.org/submit?journalID=RSOS&manu=RSOS-191863.R1>

- Competing interests

- Authors' contributions

- Acknowledgements

- Funding statement

Because the schedule for publication is very tight, it is a condition of publication that you submit the revised version of your manuscript before 11-Jul-2020. Please note that the revision deadline will expire at 00.00am on this date. If you do not think you will be able to meet this date please let me know immediately.

- 1) A text file of the manuscript (tex, txt, rtf, docx or doc), references, tables (including captions) and figure captions. Do not upload a PDF as your "Main Document".
- 2) A separate electronic file of each figure (EPS or print-quality PDF preferred (either format should be produced directly from original creation package), or original software format)
- 3) Included a 100 word media summary of your paper when requested at submission. Please ensure you have entered correct contact details (email, institution and telephone) in your user account

4) Included the raw data to support the claims made in your paper. You can either include your data as electronic supplementary material or upload to a repository and include the relevant doi within your manuscript

5) All supplementary materials accompanying an accepted article will be treated as in their final form. Note that the Royal Society will neither edit nor typeset supplementary material and it will be hosted as provided. Please ensure that the supplementary material includes the paper details where possible (authors, article title, journal name).

Kind regards,

Anita Kristiansen
Editorial Coordinator

on behalf of Marta Kwiatkowska (Subject Editor)
openscience@royalsociety.org

Associate Editor Comments to Author:

Comments to the Author:

A few minor tweaks are required but broadly the paper is on track for acceptance. Please deal with the remaining recommendations from the reviewers before resubmitting.

Reviewer comments to Author:
Reviewer: 3

Comments to the Author(s)
Thank you for your work

Reviewer: 4

Comments to the Author(s)
This is a good paper that in its revised form is much stronger than in the original one. It covers a 'hot' topic providing insightful results and contributing strongly to the literature in this field. I undoubtedly recommend publication.

Reviewer: 2

Comments to the Author(s)

All my previous comments and concerns have been sufficiently addressed in the new version. I only have a few minor comments for this revised manuscript.

1. In Abstract the authors claim to “predict the subset of the most widely adopted crypto assets and their features”. In effect, however, the authors are solely predicting characterizations (or features) of the adoptable assets, not the exact assets themselves. I suggest either (I) modifying the language in Abstract to more precisely describe the contribution; or (II) naming a few real-life examples of the adoptable crypto assets based on the authors’ prediction, to justify the contribution.
2. The authors use “distributed ledger” and “blockchain” interchangeably when the former is a broader concept than the latter. I suggest either (I) adding a brief explanation of the two terms and notifying the readers in advance of the interchangeable use of the two terms in the paper; or (II) being precise about their distinction, and modifying the sentence such as “[...] secured by cryptographic technology and shared electronically via a distributed ledger (blockchain)” into “[...] secured by cryptographic technology and shared electronically via a distributed ledger, such as a blockchain”.
3. The section Introduction has now been significantly expanded. I suggest splitting the section into two subsections for better readability: the first one on the general background of crypto investment, the second one introducing the model framework (the “app”). In addition, I believe the inclusion of Etoro screenshots from the authors’ response letter may facilitate readers’ comprehension of the imaginary app.
4. It would be ideal if the authors could make the display of the figures consistent. Specifically, for Figs 6 and 7, I suggest placing the legend for crypto types directly in the figure (as done with Fig 8), as opposed to in the figure caption. For Figs 10 and 11, legends are yet to be added.

Reviewer: 1

Comments to the Author(s)

The authors have extensively reviewed the manuscript. I judge the scope of the article is much clearer and the paper is ready for publication at this stage.

Author's Response to Decision Letter for (RSOS-191863.R1)

See Appendix B.

Decision letter (RSOS-191863.R2)

Dear Dr Bartolucci,

It is a pleasure to accept your manuscript entitled "A Model of the Optimal Selection of Crypto Assets" in its current form for publication in Royal Society Open Science. The comments of the reviewer(s) who reviewed your manuscript are included at the foot of this letter.

You can expect to receive a proof of your article in the near future. Please contact the editorial office (openscience_proofs@royalsociety.org) and the production office (openscience@royalsociety.org) to let us know if you are likely to be away from e-mail contact -- if

you are going to be away, please nominate a co-author (if available) to manage the proofing process, and ensure they are copied into your email to the journal.

on behalf of Prof Marta Kwiatkowska (Subject Editor)
openscience@royalsociety.org

Appendix A

Response to Referees' comments RSOS-191863

Dear Editor, we wish to thank the Referees for their constructive remarks that helped us produce a revised version of our paper *A model of the optimal selection of crypto assets*, which we are hereby resubmitting for publication in *Royal Society Open Science*.

We have carefully considered the Referees' useful comments and the concerns they raised about the assumptions of the model and the presentation of the results. A summary of the main changes in the manuscript following the Referees' suggestions and a point-by-point reply is attached below.

Main changes to the manuscript

- We have modified the abstract to state the objectives of the paper more clearly.
- We have modified the introduction, in particular moving the description of crypto assets classes and their classification in a separate subsection, namely Sec. 2(a)(i).
- We have modified Sec. 2 (a), (b) and (c) to clarify the main modelling choices and assumptions.
- We have added Refs. [5, 7, 8, 39, 40] and modified Ref. [25](previously Ref. [32]), to reflect the updated publication information.
- We have fixed minor typos and notation inconsistencies.
- We have repositioned Fig. 1 (now Fig. 2) and extended the captions of Fig. 1, 2, 3 and Table 1.
- We have changed the axis labels in Fig. 3 (bottom panel) to make the notation consistent throughout the paper.

Detailed response to Referee 1

We are thankful to the Referee for taking the time to carefully read our paper. We provide a detailed reply to his/her comments below.

The authors introduce an agent based model to model interactions between cryptocurrency investors. The idea of designing an agent based model where investors take decisions based on the security, stability and expected return on investments seems reasonable, although the authors do not provide any evidence that this is the case. Lack of clarity about the scope of the article. My confusion started from the third sentence of the abstract, which reads: 'Investors make choices over, crypto assets similarly to how they make choices by using a recommender app'. What does this mean?

We are sorry to read that the way we formulated our narrative induced some confusion and was not clear enough. We have clarified in the resubmitted version that the framework of the recommender app is a convenient way to explain the choices and assumptions of the model and the dynamics of adoption

and return. We have clarified this point in the abstract and in Sec. 1 and 2, by further stressing the nature and the objectives of our work.

We have also provided in Sec. 1 and Sec. 2 (c) an explicit definition of recommender apps: a recommender app or platform can be defined as a type of filtering system that aggregates information (similarly to internet search engines) and provide user-specific recommendations based on the revealed preferences of all of its users to date. We have simply extended this idea to the crypto assets case, whereby the crypto app acts as an investor-specific recommendation tool.

We would like to point out, however, that the real-life mechanism of adoption and investment in the crypto assets landscape do not differ enormously from the simplified framework that we have adopted. Information aggregators in the crypto landscape do exist and can be easily connected via APIs to apps, which investors can use to decide which asset they should invest in, based on the available information. One example that we discuss in more detail below is the *Etoro* platform.

Recently a large number of wealth management apps have also started to appear (e.g Moneyfarm, Wealthify), which collect users' preferences and information on their behaviour, and use this information (in an aggregate and anonymous form) to provide tailored recommendations and investment advice. Users of such platform willingly and knowingly decide to share some information (e.g. their preferences or adoption choices) to gain other benefits, i.e. receive recommendation that are calibrated on other users' behaviour but at the same tailored to their (risk) profile. More generally, those apps exist already for the digital marketing of non-financial digital assets, e.g. books, music or movies (see for instance Ref [8] added to the revised version for a recent review on digital marketing approaches).

As we stress in Sec. 1, crypto assets are a completely new asset class, they are digitally native and they do not have any physical presence nor counterpart. As they are so different in nature, also the preferences to invest in crypto assets may substantially differ from the drivers of adoption of standard financial assets. Similarly to other asset classes, though, individual crypto assets are held in expectation of some future economic benefits, which are often related to the aggregate return on the "market" for the asset class as a whole. We have clarified the crypto assets definition in Sec. 1 and added Refs. [5, 7]. Therefore, having a tool (such as the "crypto app") through which investors disclose their preferences, can provide a way to conveniently frame the process of "recovering investors' preferences" and then predict assets' adoption and returns.

By reading the abstract, I thought the authors had ran an experiment using an app. Instead, this work is purely numerical, but the contribution is unclear. The app designed by the authors does not exist, investors do not have global knowledge around prices and other investors choices, and they do not take decisions simultaneously. It is unlikely to imagine a scenario where that would be the case, so I am puzzled about the usefulness of these numerical analyses, which seems to be a theoretical exercise. It is important to add that although much data about cryptocurrency prices and trading are publicly available, this article never refers to or make use of actual data.

We acknowledge that our work is indeed numerical but supplemented with analytical insights [see Sec. 2 (b)(i) on the Glauber dynamics, Sec. 3 (b) on analytical estimation of the β -parameters, to mention a few examples.]. As mentioned above, in the revised version we have modified the abstract to stress the type of analysis we are conducting, and we are only introducing later on in Sec. 2 (c), the narrative device of the crypto app. The Referee raises further criticisms mentioning that:

1. *"the app designed by the authors does not exist"*. As described earlier and better explained in the resubmitted version (see Sec. 1, pag. 2 and Sec. 2(c) pag. 7), the "app framework" that we have included is just a narrative device to mimic the way investors make choices about crypto assets to invest in. This said, real-life apps/platforms with similar features as those we propose in our model do exist already on the market for different types of digitally native assets (we have discussed a few notable examples below). In the paper, we have decided not to name and single out instances of existing apps or companies providing this service, as this was not essential for

our purposes.

2. “investors do not have global knowledge around prices and other investors choices, and they do not take decisions simultaneously.” It is true that usually in real life investors may not have the full knowledge on prices and other investors’ choices, but we never assumed that this was the case in our model. The only information that is provided to each individual investor in our model is an aggregate information about the market as a whole (R_{tot}), which can be thought of as a general market trend, aggregated via the app. Indeed, only the app, equivalent to a market operator (e.g. an exchange) does have information on the actions of single investors as well as more fine-grained information on prices and returns that it uses to construct the aggregate object (R_{tot}).

We also did not assume that the investors simultaneously update their preferences, on the contrary the update rule is sequential, and after a given time frame has elapsed the global variables are recalculated. We have therefore clarified in the paper [Sec. 2 (b), (c)] that it is not down to the single investor to gain or be exposed to the full information about the prices and other investors’ choices but what they do is just to use aggregated information collected and provided by the app. Moreover, we specify that in a real setting, to protect the privacy of the users submitting their preferences, the app may be equipped with cryptographic tools to encrypt information and calculate aggregated quantities without accessing or disclosing information submitted by the individual user.

3. The Referee is right in pointing out that there is a large body of cryptocurrency trading data available for analysis. In our literature review, we indeed mentioned the relevant empirical papers that by using publicly available data have investigated similar mechanisms to those analysed in this paper, in particular the impact of adoption on returns, the “social aspects” and network effects when trading digital assets. We have also further stressed in the revised version the relevance of the empirical literature with respect to our model (Sec. 1, pag. 3).

Unfortunately, finding a one-to-one correspondence between the publicly available data and the model’s parameter is a very difficult task though. One reason is that the type of data that would be needed to validate our model are detained by private companies (e. g. trading companies or exchanges). This includes the adoption preferences and risk aversion (β parameters of the model), which cannot be retrieved from publicly available data such as market cap, price, etc. Our work could open a new avenue to explore those data, offering private companies an incentive to release anonymised and aggregated data to make predictions or validations.

Lack of clarity and realism around the model. I have found it hard to follow the text describing the model, which makes it even harder to understand the potential contribution of this article. What does adoption mean? Investments are not binary choices. Do I understand correctly that the decision of a single agent is responsible for the change in adoption concerning all users? This seems unreasonable. Why would the return depend only on the adoption probability?

We are sorry that the description was not sufficiently explicit and clear around few of the definitions and assumptions of the model. We have now updated the discussion in the paper on the points raised and we comment about them below.

Regarding the definition of adoption, we have now added a few explicitly examples in Sec. 2 (b) and clarified its definition. For instance, we have specified that in a real context the “adoption process” would correspond to exchanging Bitcoins (asset i) for Ethereum tokens (asset j). Other examples may include digitally native assets that are not tokens/coins or serve as a currency, as for example a software: a user may decide to use or invest less on software A compared to software B , or download from the Google play store “app A” and delete from their devices “app B”.

We have also clarified the meaning of the update of adoption probabilities in Eq. 2.2. First, in the revised version we have written Eq. 2.2 in the most general form, to include time variability and other

factors the jump in probability of adoption may depend on. Then, we have also explicitly clarified the meaning of the increment δ , which depends on network externalities, i.e. number of other users u_i possessing or trading a given asset i . In the context of digitally native assets, popularity indicators are, indeed, readily available to investors and users, and they are explicitly advertised with the aim of attracting attention on the digital platform or asset. Examples include number of downloads of a digital object (a file, an app), number of accesses or users of a digital platform, number of wallets on payment platforms, number of transactions issued.

Regarding adoption/investment choices and the binary modelling choice, there are many examples in the context of digitally native assets where adoption choices can be well approximated as binary choices: if one considers the example of exchanging one crypto asset for another one (e.g. Bitcoin for Ethereum), using a wallet application or through an exchange, the investor is indeed making a binary choice. This said, our simplified modelling assumption can be easily relaxed to include more sophisticated scenarios. For instance, within the current framework, we could easily consider the case where an investor compares asset i against the ‘‘average features’’ of all other assets. Those changes would not drastically impact the analysis nor the results. We have now clarified this point in Sec. 2 (b) as well.

Regarding the effect that a single agent may have on the decisions of all others, in our model every user independently submits their adoption choice. The recommendation on whether the proposed choice is a sensible one, is formulated using information on the future economic benefit of the assets as well as the *aggregate* behaviour of other users. As mentioned above, in a digital setting information about users’ behaviour and adoption choices are often publicly available and easily accessible and/or aggregate via apps or platforms and provided to interested users/investors. Good examples are wealth management apps, as for instance *Moneyfarm* (<https://www.moneyfarm.com/uk/investment-advice/>), where users answer questionnaires to identify their attitude to risk, investment requirements and financial history, and the ‘‘robo-advisor’’ service provides advice on digital investments. Another interesting example is the platform *Etoro* (<https://www.etoro.com>), where users can trade currency pairs, indices and commodities via the platform: on this app users, once they sign up for the service, can literally ‘‘copy’’ the actions and portfolio of other users of the platform (see Fig. 1).

Figure 1: Screenshot of the Etoro platform and the functionality ‘‘copy people’’ where users can copy portfolio and investment choices of other investors in the platform.

The return does not depend only on the adoption probability. As highlighted in Eq. 2.7 the dynamics of the return also depends on the intrinsic assets’ features (specifically on the stability of the asset) via the noise term $\eta(t)$.

Detailed response to Referee 2

We wish to thank the Referee for the detailed report and comments on our manuscript. We provide below our detailed response to the issues raised.

Using security and stability as the only two dimensions of a crypto assets simplifies the modelling, but needs to be better motivated to make the model more realistic and applicable. Importantly, the authors omit the fact that many crypto assets, especially those categorized as “crypto tokens” in the paper, are designed to have specific utilities targeting different user cohorts. That is to say, depending on their view on the “business model” of a crypto token, different users can score the same crypto token differently, even if they value “security (technology)” the same way. Page 3 lines 32-38: the authors categorize “crypto tokens” as having “low security”, but did not demonstrate why that’s the case in the description of crypto tokens.

In our framework, we define security and stability as *intrinsic* features of the token: the security of the token depends on the cryptographic primitives it is based on, not on the perception one has of the security of the tokens. The “score” of the users, i.e. their perception of stability and security, is instead modelled and taken into account via the β -parameters in the model (introduced in Sec. 2(c) (iii)). In the revised section, we have added a further example based on the Referee’s comment in Sec. 2 (c)(iii).

Regarding our categorisation of crypto assets (low/high security, low/high stability), in the revised version we have improved the description of the criteria according to which we classify the assets (see Sec. 2(a)) and we have provided few specific examples and reasons behind the classification (Sec. 2(a)(i)). The fact that crypto tokens, in particular, are categorised as “low security” is mainly related to the number of attacks aimed at stealing funds they suffered (see added Reference [51] for a summary of such attacks). Indeed, despite having characteristics similar to cryptocurrencies (e.g. decentralisation and consensus protocol), they are often issued by smaller communities, who do not always disclose information about the cryptographic tools deployed, and are held in less secure wallets. This said, we have also clarified that this simplified categorisation is provided as a general guidance on the different existing types of assets but indeed in reality the situation is more nuanced.

While it is nice and helpful to describe four categories of crypto assets, the description in its current is way too long and not so essential for the model.

We have carefully considered this remark, also in light of the other Referees’ comments, and we believe that keeping this description is not harmful to the reader. All the results are conveniently summarised in terms of the behaviour of the four macro asset classes and we are worried that without providing a definition of what those classes actually represent a reader, who is approaching this research strand for the first time, would not be able to follow through the paper. We have, though, moved the description of the assets categories within a separate subsection (see Sec. 2(a)(i)) to improve the readability of Sec. 1.

It is slightly confusing to categorize crypto assets with two binary features — high or low security, and high or low stability — and later in the model, those two features appear to be continuous.

The macro binary classification was added mostly for expositional purposes. In order to avoid any further confusion, we have clarified the binary vs. continuous features issue in Sec. 1, 2 (a), 2(b) and in the caption of Table 1. We state explicitly that the range of the features is continuous, but that we use the macro (binary) classification to simplify the description and provide the reader with a working example of a possible classification of existing crypto assets.

Moreover, we state that the simplified binary classification framework would yield qualitatively the same results compared to the scenario where security and stability are assumed to be continuous and we stress that the continuous modelling allows for accommodating within-class heterogeneity. For instance, if we consider the stablecoins case, depending on the collateralisation method they are based on they may be classified with very heterogeneous “stability” and “security” parameter values, despite

belonging to the same macro crypto asset class (low security [$s < 0.5$]/high stability [$\xi \geq 0.5$]).

Page 4, line 23-31: the description of sentiment analysis appears too verbose and strays off the central theme.

We have reduced and tightened the paragraph on the literature on sentiment analysis, but we have added an explanation on why this strand of literature is deemed to be relevant in our context. Indeed, our model focuses on the interplay between network effects, users' adoption and future expected value of the crypto assets. The literature strand that we have mentioned provides empirical attempts aimed at quantifying this feedback loop between adoption and return and its drivers.

The placement of Figure 1 is currently too early — many notations and variables are not explained in the preceding text. I recommend to also explain all the notations either directly in the figure, or in figure notes. More elaborate figure notes will be appreciated for other figures as well.

We thank the Referee for this remark. We have moved Fig. 1 (now Fig. 2 in the revised version) at the end of Section 2, to serve as an overall summary of the different stages of the dynamics described within Sec. 2. We have also extended the caption to include a more detailed explanation of the notation and the meaning of the figure. We have also extended and/or clarified the captions of Fig. 1, 3 and Table 1 as per the Referee's suggestion.

There are numerous instances of notation clashes in the paper that hampers the understanding. For example, “k” in Eq. 2.1. stands for whether a token is adopted or not, while on page 15 line 26 it stands for an asset, and on page 16 line 34 it stands for an investor. Another example, “j” in Eq. 2.2 is one of the assets in an asset pair, while on page 6 line 13, “j” is a counter.

We thank the Referee for flagging those inconsistencies that may generate some confusion. We have now fixed all of them in the revised version.

Detailed response to Referee 3

We thank the Referee for critically reading our manuscript and for providing a detailed report. We are very pleased to read that he/she found that the paper was clearly presented and that the analyses and results were well supported.

The conception behind cryptocurrencies was to avoid any central banks regulations and to “give control of the money” to the people. So the CBDCs may not be seen as stable from the perspective of traditional cryptocurrency community, because central banks can easily change the supply of the currency. In the case of Bitcoin and many others cryptoassets it is fixed at some maximal level. So in my opinion the authors should more precisely describe selected stability criteria and emphasized the fact that they may be not commonly accepted.

We thank the Referee for this remark. In light of this comment we have clarified the discussion around the stability and security criteria in Sec. 1 and Sec. 2(a). We have also added two further references (Ref [39,40]) to better define the assets' features and possible types of vulnerabilities. We have also stressed that we are aware of the limitations of our classification, which in the context of our paper mostly serves as an illustration (Sec. 2(a)(i)).

The variables in Fig 3: X_{cm} and Y_{cm} are not explicitly defined.

We thank the Referee for pointing this out. We have now changed the notation in Fig. 3, using instead $1 - \bar{a} = X_{cm}$, $\bar{r} = Y_{cm}$ (see Eq. 3.1) to introduce the “centre of mass”.

Line 56 page 13 space missing “assets(high...”

Ref 32 was already published - see <https://www.mdpi.com/1999-5903/11/7/154>

We have fixed the typo and updated the reference to reflect the most recent published version.

We trust that our revision has addressed carefully all Referees' remarks and we hope that the paper will now be found suitable for publication in *Royal Society Open Science*.

Yours sincerely,

Silvia Bartolucci & Andrei Kirilenko

Appendix B

Response to Referees' comments RSOS-191863.R1

Dear Editor,

we wish to thank the Referees for their further constructive remarks on the revised version of our manuscript *A model of the optimal selection of crypto assets*.

The revised version was reviewed by four Referees. Three of them recommended acceptance, while Reviewer 2 offered some further minor suggestions, which we have all incorporated in this newer version. More precisely:

1. *In Abstract the authors claim to “predict the subset of the most widely adopted crypto assets and their features”. In effect, however, the authors are solely predicting characterizations (or features) of the adoptable assets, not the exact assets themselves. I suggest either (I) modifying the language in Abstract to more precisely describe the contribution; or (II) naming a few real-life examples of the adoptable crypto assets based on the authors’ prediction, to justify the contribution.*

We thank the Referee for suggesting that we should be more precise in the abstract. We have reformulated the second sentence in the abstract as: Under a variety of modelling scenarios – e.g. in terms of composition of the crypto assets landscape and investors’ preferences – we are able to predict the features of the assets that will be most likely adopted, which can be mapped to macro-classes of existing crypto-assets (stablecoins, crypto tokens, central bank digital currencies, and cryptocurrencies).

2. *The authors use “distributed ledger” and “blockchain” interchangeably when the former is a broader concept than the latter. I suggest either (I) adding a brief explanation of the two terms and notifying the readers in advance of the interchangeable use of the two terms in the paper; or (II) being precise about their distinction, and modifying the sentence such as “[...] secured by cryptographic technology and shared electronically via a distributed ledger (blockchain)” into “[...] secured by cryptographic technology and shared electronically via a distributed ledger, such as a blockchain”.*

We thank the Referee for suggesting that we should be more precise and consistent in the use of the terms “blockchain” and “distributed ledger” throughout the paper. We have modified the first sentence of the Introduction as: A crypto asset is an intangible digital asset whose issuance, sale or transfer are secured by cryptographic technology and shared electronically via a distributed ledger. A distributed ledger, in turn, is a database of issuance and transaction records (ledger), copies of which are stored on multiple computing devices (nodes) that form a distributed computer network. A well-known instance of a distributed ledger is the *blockchain*, where information about transactions is stored in a characteristic format consisting of blocks of data chained together via cryptographic primitives. In the following, we will use the two terms “blockchain” and “distributed ledger” interchangeably for simplicity.

3. *The section Introduction has now been significantly expanded. I suggest splitting the section into two subsections for better readability: the first one on the general background of crypto investment, the second one introducing the model framework (the “app”). In addition, I believe the inclusion of Etoro screenshots from the authors’ response letter may facilitate readers’ comprehension of the imaginary app.*

We thank the Referee for his/her suggestions. We have taken it on board, and split the Introduction separating the general introduction from the “Related Works” subsection, which is devoted to a thorough literature review. On pag. 3, we have added the following description of Moneyfarm and Etoro as prominent examples of platforms that would fit within the modeling framework proposed in our paper: In the context of crypto-investments and wealth management, for instance, interesting examples are provided by the following platforms. On *Moneyfarm* (<https://www.moneyfarm.com/uk/investment-advice/>), users answer questionnaires to identify their attitude to risk, investment requirements and financial history, and the “robo-advisor” service provides advice on digital investments. On *Etoro* (<https://www.etoro.com>), users can trade currency pairs, indices and commodities via the platform: on this app, individuals can also peek into other users’ profiles and literally “copy” their actions, portfolio and preferences and embed them into their own investment strategy. However, we do not feel comfortable adding screenshots of these platforms webpages to the paper because of copyright issues, as our aim is not to advertise any specific commercial application, but rather to give the reader a flavor of the existing landscape of platforms and crypto-apps.

4. *It would be ideal if the authors could make the display of the figures consistent. Specifically, for Figs 6 and 7, I suggest placing the legend for crypto types directly in the figure (as done with Fig 8), as opposed to in the figure caption. For Figs 10 and 11, legends are yet to be added.*

We have standardized the labels and included legends directly in all relevant figures of the paper as suggested.

We trust that our revision has addressed carefully all Referees’ remarks and we hope that the paper will now be found suitable for publication in *Royal Society Open Science*.

Yours sincerely,

Silvia Bartolucci & Andrei Kirilenko